
**Research article: Household resilience to major slow kinetics floods: a prospective**
**survey of the capacity to evacuate in high rise buildings in Paris**
**Nathalie Rabemalanto [1], Nathalie Pottier [1, *], Abla Mimi Edjossan-Sossou [2, 3], Marc Vuillet [3]**
*Correspondence to : Nathalie Pottier (nathalie.pottier@uvsq.fr)*
[1]   nath.rabe@gmail.com, nathalie.pottier@uvsq.fr. Center for Studies on Globalization, Conflicts,

8        Territories and Vulnerabilities (EA4457 CEMOTEV-UVSQ), University of Versailles Saint Quentin-

9        en-Yvelines - Paris-Saclay, France.

[2]   medjossan@gmail.com . University of Lorraine, CNRS, CREGU, GeoRessources laboratory, Nancy

11       School of Mines, Campus Artem, CS 14234, Nancy Cedex, F-54042, France

[3]   marc.vuillet@eivp-paris.fr. Lab'urba, University Gustave Eiffel, Paris School of Urban Engineering,

13       France

* *Corresponding author:* nathalie.pottier@uvsq.fr
**Abstract:** This article presents the results of a prospective survey of households living in the only high
rise residential buildings of Paris, which are located in a flood zone. It questions the behavior of
households likely to be subject to evacuation instructions in the event of a progressive flooding impacting
the functioning of the technical networks and associated urban services. The survey received 523
responses from 11 residential high-rise buildings located in the 15th district of Paris. It assessed the
propensity of households to evacuate autonomously through three main factors: the capacity to self-
evacuate, to self-host and to go to this temporary accommodation. The survey answers explicit requests
for information by local authorities on inhabitants' capacities to self-evacuate and to self-host in order to
support the formers' estimation of shelter requirements. The typology of evacuation capacities reveals
that most of the respondents are partially dependent due to difficulties relating to re-accommodation
issues. Furthermore, many people seems to have an incorrect perception of the public authorities'
responsibilities. Information and warning systems could thus be improved, notably through a participative
method.
**Keywords:** flood, evacuation, household resilience, prospective survey, Paris.



## 1. Introduction

A major flood of the Seine in Paris area would be a terrible challenge for crisis management services,
inhabitants and the economy of affected territories, regardless of whether they are directly affected by
flooding or not. According to the OECD (2014; 2018), a flood with a water level similar to the 100-year
flood of 1910 would directly affect 1,000,000 people, with a flood duration of about one month. Nearly
2,000,000 customers would be without electricity and nearly 5,000,000 without water. A very large number
of people would therefore be heavily impacted without for all that suffering the direct impacts of the flood
itself.
Various protection systems, including mobile or more conventional levees, have been designed to
limit the extent of flooding (OECD, 2014). Nevertheless, their effects appear to be highly uncertain, mainly
because of the unknowns of the risk of groundwater levels rising or the failure of a levee/cofferdam (Gache,
2014). As a result of this, many technical networks and urban services would be shut down as a preventive
measure. During the flood of May-June 2016, we witnessed the shutdown of the regional express train
(RER C), which carries nearly 550,000 passengers a day, numerous power cuts and the evacuation of
nearly 20,000 people. This flood, which was serious on a number of modest tributaries of the Seine (Loing,
Yvette, Essonne in particular), remained a phenomenon of low amplitude within the Ile de France region,
being considered as a 20-year flood in the city of Paris.
The risk of a major flooding of the River Seine would primarily raise the question of the fate of the
830,000 people living inside the flood zone (OCDE 2014), compounded by the numerous people indirectly
affected (power cuts, water and/or sanitation supply disruption, etc.). People who might have to evacuate
should be cared for or be able to relocate for a period of days or even weeks, anticipating the kinetics of
the flood. In this paper, we investigate the capacity of inhabitants living in the densely populated areas of
the Paris urban area to self-evacuate and self-relocate in the event of a major flood of the River Seine.
Kolen (2013) highlights the complexity of evacuation issues for large populations, stating that "*as the size*
*of an evacuation increases, its complexity also increases*". In the present study, not only is the population
size large compared to the small area to be evacuated (cf. presentation of the survey area below), but the
height of the buildings in question exacerbates the complexity of the evacuation process. When would the
residents leave, knowing that the feeling of security in high-rise buildings might not favor the decision to
evacuate? Which household profiles are likely to leave first? What are the factors which facilitate or
handicap the autonomy of the households in the event of evacuation? These are just some of the issues that
this case study raises.
Several researchers have studied the management of a major flood of the Seine in the Ile de France
region. These studies examined the issue from a global standpoint (Reghezza, 2006) and from the point of
view of the crisis management by national and regional services (November & Créton-Cazanave, 2017).


They also relate to the continued activity of network operators and urban services (Toubin *et al.*, 2015;
Bocquentin *et al.*, 2020), the mobility and reassignment of employees who can no longer go to their
workplaces (Lhomme *et al.*, 2019), social impacts (Fujiki & Renard, 2018) and household evacuation
factors (Fujiki, 2017). Based on the cartographic exploitation of statistical indices and a bibliographical
study, the work of Fujiki (2017) adopted a global approach to estimate the number of households that
would need to be relocated for several major flood scenarios in the Ile de France region. This work
represents a major breakthrough, making it possible to determine orders of magnitude for evacuation rates
and evacuees requiring rehousing. Nevertheless, several additional pieces of data could usefully refine and
supplement these results, in particular those relating to the inhabitants' perception (Navarro *et al.*, 2016)
of the risk and the precautionary actions (Grothmann & Reusswig, 2006) as well as of the brakes and assets
relating to self-evacuation and to self-hosting.

In this research, we propose to assess the household resilience in the face of an evacuation caused by
a major flooding of the Seine, using a prospective survey. The aim is to try to identify the self-evacuation
and self-relocation capacities of people living in a very high-density neighborhood, such as the
Beaugrenelle high-rise flats located in the 15th district of Paris, in the face of a slow-motion flood scenario.
We try to answer the following questions:
•    What are the predominant factors influencing the target households' decision to evacuate?
•    What is their perception of the risk?
•    Do they have a means of travel and relocation?
•    Are they able to continue their professional activity from their temporary place of residence?

The database used for this study is that of a prospective questionnaire conducted in 11 high-rise
buildings in Paris. They are located in the 15th district, in an area along the banks of the River Seine. The
data is provided by 523 respondents, representing 23% of the total number of residents who received the
questionnaire. There are only a few residential high-rise buildings in Paris. The presence of this type of
building in the "Front de Seine" zone has made it the most densely-populated area in the immediate
vicinity. It is also more highly exposed to flooding, as demonstrated in the Flood Risk Prevention Plan
(DULE, 2007). The survey explored the extent to which the residents are able to self-host and, to a slightly
lesser extent, to self-evacuate. It also aimed to help determine the factors which lead to evacuation.
The remainder of this paper is structured as follows. First, the factors that can influence households'
decision to evacuate in response to a natural disaster are presented. The equipment and methods used for
the survey are then described together with an analysis of the results. The literature on evacuation decision-
making justifies the content of the questionnaire. The results section will then illustrate the global trends
relating to the characteristics of the sample, the constraints and factual information concerning the
respondents' capacities and their perceptions of flood risk and evacuation. In large part, the results will
highlight a typology corresponding to the propensity to evacuate. Finally, the respondents express their


expectations regarding the transmission of information and the evacuation process. These suggestions have
been classified in order to help the authorities and everyone involved to define their strategies and actions
when preparing the evacuation. The conclusion emphasizes the contributions of this study and highlights
new avenues for reflection.

## 2. Factors influencing a household's decision to evacuate in the face of natural disaster

The factors which lead households to decide whether or not to evacuate in situations of risk could be
of an intrinsic and extrinsic nature. Among other things, these factors involve a household's capacity-
related factors, risk perception, the structural and functional inhabitability of the place of residence, social
influence and environmental factors facilitating or hindering the possibility of evacuating (Mileti, 1995;
Dash & Gladwin, 2007; Lim *et al*., 2016; Ahsan *et al*., 2016).
Evidence exists of correlations between households' socio-demographic characteristics and their
ability to leave or to stay in an area threatened by a hazard (Parker *et al*., 2009). Generalizing these factors
could nevertheless be problematic because the correlation can be negated or even reversed according to
the case in question. Depending on the specific context of the area studied, the socio-demographic
characteristics underlying a household's ability to evacuate may include, but are not limited to, gender
(Mileti, 1995; Fraser *et al*., 2014; Luathep *et al*., 2013), household size (Luathep *et al*., 2013; Smith &
McCarty, 2009), the presence of vulnerable people such as children, senior citizens or persons with
disabilities (Luathep *et al*., 2013; Lim *et al*., 2016), ownership of and access to a vehicle (Wright &
Johnston, 2010; Luathep *et al*., 2013), access to an available relocation place (Chang *et al*., 2009), the
presence of pets (Drabek, 2001; Heath *et al*., 2001a, Solis *et al*., 2010), etc. Because these factors vary
from one household to another and the significance of their influence also varies depending on the context
(Murray-Tuite & Wolshon, 2013), identifying households likely to evacuate can prove complex (Wright
& Johnston, 2010).
Apart from socio-demographic characteristics, a household's intrinsic factors that can lead it to
evacuate may include risk perception (Solis *et al*., 2009): people can make an appropriate evacuation
decision if they are aware of and understand their risk level (Piatyszek & Karagiannis, 2012). According
to Jumadi *et al*. (2018), risk perception can be understood as the way households interpret the likelihood
of threat; some households may consider themselves to be safe, thereby tending to think that evacuation is
not necessary. A household's risk perception, and consequently its decision to leave or to stay, depends
mainly on its previous experience of disasters (Dash & Gladwin, 2007) or its risk awareness (Whitehead
*et al*., 2000).
A household's behavior in the face of disasters also depends on certain extrinsic factors such as
communication and information concerning the risk (De Jong & Helsloot, 2010). Households may decide
to evacuate if they hear appropriate emergency information. Furthermore, in the face of natural disasters,
people may decide to leave due to the inhabitability of their residence on the grounds of safety, utilities





shut-off and health (Wright & Johnston, 2010). Residents may indeed evacuate if they deem that the level
of damage to their home caused by the hazard is so great that remaining inside could be unsafe or their
well-being could also be affected. They might therefore leave their home when facing a disruption of
lifelines provided by technical networks, including power outages, urban heating shut-offs or water supply
system failures (Chatterjee & Mozumder, 2015). Furthermore, as social beings, a household's decision
could be influenced by the society in which they live. They may take a decision based solely on their
individual convictions and capacities or they might follow the example of their neighbors after seeing them
evacuate (Lindell *et al*., 2005; Jumadi *et al*., 2018). Environmental cues may, for example, include hazard-
related factors like sights, sounds or smells that indicate the onset of disaster, or the distance from the
source of the hazard (Smith & McCarty, 2009; Lindell *et al*., 2015). This type of cue also involves the
"livability" of a household's neighborhood. The loss of normal operation of support systems and services
(public transport, businesses, etc.) required to ensure a household's well-being and functioning may make
it difficult to remain in their home (Wright & Johnston, 2010).
This study will mainly focus on intrinsic factors of the targeted households to gain an improved
understanding of their capacity to self-evacuate, to self-host, and to move to a relocation place. This will
help defining a typology of evacuation propensity that could be used to support the design of efficient
evacuation strategies.

**3. Methodology: A prospective survey on household evacuation capacities**
***3.1. The specificities of the study area include high-rise buildings exposed to the risk of flooding***
If we only consider the 20th and 21st centuries, the most extensive flooding of the Seine in Paris
occurred in 1910. Despite the dams and levees that have been erected, the flood risk remains, even within
the most densely populated neighborhoods of central Paris, as shown on the map (Fig. 1). This map shows
the areas in the 15th district liable to flooding. In reality, there is little chance that the water would reach
street level. However, water could penetrate underground car parks, mainly by dynamic capillary rise in
the foundation walls. The actual issue in such an area is rather that technical network operators would have
to implement preventive actions by disrupting the services. This raises the temporality issue of evacuation,
as people would not see water in the streets or their buildings, but might have to leave because of the
disrupted services.

Natural Hazards
and Earth System
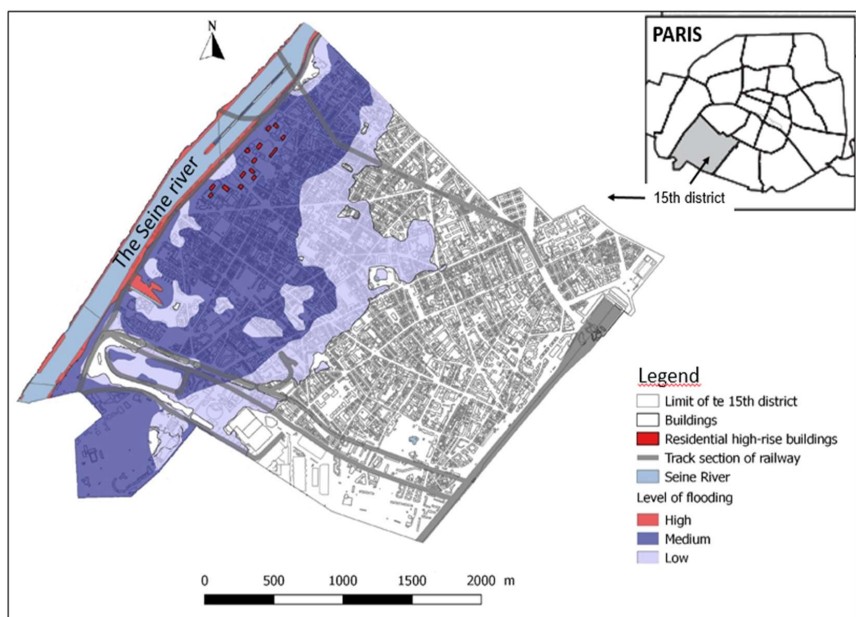


**Fig. 1.** Flood risk zoning in the 15ᵗʰ district of Paris (Source: data from the Regional and
Interdepartmental Office of Energy and the Environment, mapping by N. Rabemalanto and N.
Pottier).


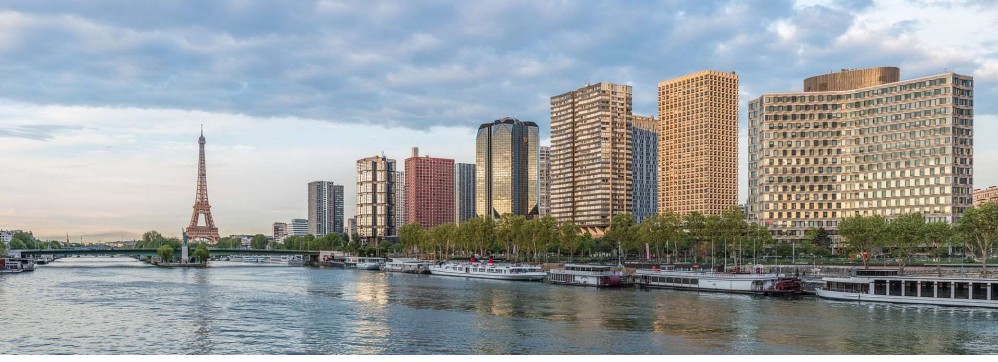


**Fig.2.** The residential high rise buildings of the Front of the Seine river in the 15ᵗʰ district in Paris
(source: https://en.wikipedia.org/wiki/Front_de_Seine).






The 15th district was chosen for this study because it is widely exposed to the risk of flooding and is
the most densely populated district in Paris (INSEE, 2016), due to the existence of residential high-rise
buildings located exclusively in this territory along the Seine (fig.2). In 2015, the number of inhabitants in
this district was nearly 234,000 while the density in the district has been quite stable since 1968 at around
28,000 inhabitants/Km² compared to 21,000 for Paris as a whole (INSEE, 2016). Not only is this district
the most densely populated because of the residential high-rise buildings, but the economic stakes in this
area are also highly important. One of the biggest shopping malls in Paris is located here. Moreover, some
of the high-rise buildings located in the "Front de Seine" area house companies or short- and mid-term-
stay hotel residences. It is worth noting that this applied study examines the evacuation of the residential
high-rise buildings only, rather than shopping mall visitors and hotel customers. This is because the
residents are necessarily concerned with evacuation in the event of slow-kinetics flooding, and this would
influence evacuation decision making.
Most of the residential high-rise buildings are built on an area 1 Km long (0.62 miles) and 200 m wide
(218 yds). They have four levels of parking lots, two of which are at -2 and -1 in relation to street level.
The car parks must therefore be evacuated even before the residents. This makes it more complex to
coordinate the information concerning the evacuation of residents and cars. Another crucial piece of
information is that the electrical systems of many of the buildings are located either at level -2 or -1. The
buildings concerned are therefore vulnerable even before the Seine overflows its banks due to rising water
in the basement. To limit damage, preventive power cuts inside these buildings can be implemented by
operators several days before the water invades the streets. Evacuation is therefore mandatory since it
involves the shutdown of the elevators and the height of the buildings makes it impossible to keep people
inside. If some residents still choose to stay despite being advised to evacuate, mobility would be essential,
especially for those living on upper floors.
Moreover, these people increase their exposure to other risks likely to cause domino effects which
would amplify the disaster, such as the risk of fire and the impossibility for firefighters to intervene quickly
to rescue those who have remained at home. In this case, slow kinetics flooding that does not cause death
in the Paris region can turn into a deadly risk in high-rise buildings that have not been emptied of their
inhabitants. Evacuation is therefore critical in the case of high-rise buildings in order to safeguard people's
lives and their access to all basic services. Several authors provide a clear explanation of what critical
networks are and the different ways whereby they can be interdependent. Using tangible examples, they
show how network disruptions can exacerbate crisis considerably (Toubin *et al*., 2015; OECD, 2014). For
all these reasons, preventive evacuation must be encouraged.

### 3.2. Questionnaire design

Data for this study was collected by means of a self-administered questionnaire (see in appendix). The
questionnaire was entitled: "Are you prepared for the evacuation of the Front de Seine towers?". It was



designed to gather data on household intentions regarding an autonomous evacuation (that is to evacuate
or to remain at home) and the availability of evacuation destinations as well as modes of self-travel in the
case of major flooding of the River Seine.
Even at the international level, there were only a few surveys on preparation for evacuation and
decision making in the event of flooding with slow kinetics (Fujiki, 2017). Becerra *et al.* (2013) asserted,
however, that when a hazard is weak, vulnerability is also weakened. Often, the existing surveys deal with
the case of hurricanes, tsunamis or earthquakes (fast kinetics). For instance, many research works have
made a significant contribution to the progress of knowledge about evacuation in the case of hurricanes
(Huang *et al.* 2012; Dash & Gladwin, 2007). They found that the characteristics of the hazard were the
main factor in determining evacuation decision-making (Whitehead *et al.*, 2000; Whitehead, 2005; Huang
*et al.*, 2012).
As for the type of survey, at least since the 1950s, researchers have been interested in people's
responses to risk (Baker, 1991; Thompson *et al.*, 2017), but most of the existing analyses on evacuation
behavior focus on populations that have already experienced the situation (retrospective surveys). Some
of the most well-known papers are those of Baker, 1991; Dash & Gladwin, 2007; Dow & Cutter, 2000;
Gladwin *et al.*, 2001; Zaalberg *et al.*, 2009. Some more recent papers also used retrospective surveys,
notably Demuth *et al.*, 2016; Lindell *et al.*, 2019; Wallace *et al.*, 2016. There are relatively few papers on
prospective surveys examining the intention of households to evacuate following a disaster (Fraser *et al.*,
2013; Lazo *et al.*, 2015). The challenge for this study in a Parisian district is thus its prospective
characteristics. The prospective method is much more common in the fields of medicine, management,
psychology, etc. Nevertheless, papers presenting evacuation modelling are also qualified as prospective
studies (see for example Gladwin *et al*, 2001) as they aim to predict what would happen based on the
context and the assumptions. Instead of using random parameters as in the modelling process, this paper
relies on respondents' declarations to provide an initial vision of people's perceptions, capacities and
willingness to evacuate through a qualitative method.
The key questions for the analysis of evacuation conditions were inspired by decision models found
in the literature. One of these is the Protective-Action Decision Model (PADM; Lindell & Perry, 1992,
2012), which summarizes very well the different factors influencing the psychological processes of
evacuation decision-making. It analyses the environmental and social cues, the information process and
devices (sources, information channel access and preference, warning messages) and the receiver
characteristics (Huang *et al.*, 2012).
In our survey, the questionnaire contains 23 questions with the following groups of variables (these
groups of variables do not detail expressly every question asked in the questionnaire. The latter is available
in the appendix). All questions asked were closed, except two questions on the respondents' expectations
regarding the evacuation process and the information related to it.


• Respondents owning pets and difficulties in transporting them: pets might hinder the evacuation
process mainly because their transportation might delay or make the departure more complex (Heath
*et al*., 2001b).
• The level of car park, if the respondent has one: the evacuation issue can vary according to the level
at which the respondent's car is parked. First, those with a car parked at level -2 or -1 are more likely
to be obliged to move it away if needed. Second, receiving an evacuation order for the car park might
incite them to prepare themselves to evacuate soon as well.
• Knowledge about some basic information and the perceptions on the flood risk and evacuation
process: this relationship between risk perception and the adoption of preventive behaviors is treated
extensively in the literature (see, for example, Peretti-Watel, 2000; Becerra *et al*., 2013).
• The main possible reason for evacuating: the respondent has to choose from the different reasons
suggested (cf. questionnaire in *appendix*). The study might have revealed reasons linked to the fact
that the respondents live in high-rise buildings. However, the impact of living in a high-rise building
on their answers could not be verified as no direct questions were asked about this matter. A
comparison with the reasons for evacuating identified in the literature in other contexts can
nevertheless help to verify whether or not living in a high-rise building has any influence on the
answers provided. Furthermore, this variable indicates the proportion of people who would be
sensitive to evacuation advice and orders from public officials. Many studies have confirmed that the
type of dwelling strongly affects household evacuation (Baker, 1991; Gladwin & Peacock, 1997;
Horney *et al*., 2010; Huang *et al*., 2012; Lindell *et al*., 2005; Whitehead, 2005; Wilmot & Mei, 2004;
Zhang *et al*., 2004). One might also consider that predicting the reason for evacuating automatically
also makes it possible to predict the timing of people's departure. the former variable (the reason for
evacuating) must be distinguished from the departure timing, according to past findings (Huang *et al*.,
2012; Lindell *et al*., 2005).
• The existence of a relocation destination and the possibility of continuing going to work or working
at that place: law n° 2004-811 of August 13, 2004 on the Modernization of Civil Security recommends
that people self-evacuate and self-host. This is why people are asked if they have a place to which
they can relocate and if they can get there themselves. This law postulates that people should not count
solely on public authorities in the event of an evacuation. It states that citizens must be responsible
for their own safety. Accordingly, they must have a place to which they can relocate. Furthermore,
the impossibility of continuing going to work or working at the relocation site can provide a reason
not to evacuate. This question is therefore important when wanting to assess the proportion of people
who would be willing to evacuate. Moreover, people are given the possibility in our questionnaire of
specifying where their relocation site is. Sometimes, this makes them directly determine who would
host them and whether they expect assistance from other people (public authorities, family, friends,
etc.) or whether they would just not go to that site. This is what some authors call the effect of social





cues, meaning that during the evacuation decision-making process, people expect to receive help from
others (Dash & Gladwin, 2007; Huang *et al*., 2012).
• The expectations regarding the evacuation process and the information related to it: as the respondents
could not express themselves broadly throughout the questionnaire, two questions allow them to do
so here. They have the opportunity to write short texts, which might relate to some tangible actions
they expect to be taken or how they would like to be better informed about the risk and evacuation
process. They may also specify certain information they need in order to better prepare themselves
for the hazard and for a potential evacuation.
• The characteristics of the respondent and their household: the socio-demographic variables are
systematically analyzed when conducting a study about evacuation. Many authors (for instance Alou,
2018; D'Ercole, 1991; Ruin *et al*., 2008; Villa & Bélanger, 2012) have highlighted the fact that socio-
demographic characteristics influence the way people face a hazard. Nevertheless, some authors (such
as Baker, 1991; Dow & Cutter, 1998; Huang *et al*., 2016) found in case studies that socio-demographic
characteristics were not significant factors of the decision to evacuate. As Murray-Tuite & Wolshon
(2013) stated, the significance of these characteristics in influencing evacuation decisions varies
according to the context.

### *3.3. Data collection and difficulties in accessing highly-protected buildings*

The printed questionnaires were distributed and collected over a 12-week period in spring and summer
2019 by a postdoctoral fellow, helped on certain days by several others postdoctoral fellows and
researchers. This period was chosen on practical grounds relating to the start of the survey. The
particularity of this survey was that there could be no direct interaction between the investigator and the
respondents. In fact, most of the buildings included luxury residences. Security measures and privacy
considerations made it impossible to conduct a face-to-face survey. Consequently, the survey was based
on voluntary sampling as the residents received the questionnaires and could choose whether or not to
respond. The study area comprised 14 residential high-rise buildings. As the trustees of three of them did
not allow the access to their buildings, the data were drawn from 11 buildings.
To prepare the survey, the lessors or trustees had to be informed and most of them helped organize
the distribution process by asking the building managers to cooperate with the research project team. The
term "manager" is used throughout this paper in order to facilitate reading, although some of them are
concierges and do not have exactly the same functions as the building managers. One of two methods of
distributing the questionnaire was adopted, depending on what best suited the building managers and the
organization of the each building: some were left in the mailboxes while others were left at the building's
reception desk. Distribution via the mailboxes proved to be slightly more successful, as long as the building
manager helped convince the residents to respond. Residents could leave the completed questionnaire at



the reception desk or return it by post. In one of the buildings, all respondents were obliged to return it by
post in a pre-stamped envelope, as there was no reception desk in the building foyer.
With a total of 523 respondents and over 2,283 questionnaires distributed, the response rate was 23%.
In light of the difficulty encountered in accessing these highly-protected buildings, the survey period (with
many households already on vacation) and the fact that a lot of people in these buildings were foreigners
often travelling for months at a time (according to the building managers), this rate is quite acceptable for
voluntary participation. Only three buildings displayed a response rate of less than 20%. Accordingly,
almost one in four people per building answered the questionnaire. However, voluntary response means
that sampling might be biased as only those people already aware of or curious about the topic may have
responded. It is important to take this into account because the survey itself concerns the willingness to
evacuate. If a person were not willing to evacuate and thus refused to answer the questionnaire, this would
represent a considerable loss of information. The present results nevertheless remain valid even though
they do not necessarily represent everyone's situation and opinion. In comparison, the following response
rates are those of evacuation surveys with people who have actually experienced a catastrophe (cited by
Huang *et al*., 2012): 25.7% for Hurricane Bret, 24.6% for Texas coastal evacuation expectations, 33.5%
for Hurricane Katrina, and 35.6% for Hurricane Rita. The present study, however, concerns a hypothetical
event that has not been experienced. People might be more willing to respond to a survey about their actual
experiences, so this 23% rate for a prospective survey is relatively acceptable.

*3.4. Analysis method: typology of households according to the level of autonomy in an evacuation*
*situation*

The main results will be provided in the form of a households's typology expressing their level of
autonomy in the event of evacuation. The following five criteria are used to produce it:
• C1: intention to evacuate relying on stated reasons, bearing in mind that some people will not
evacuate, regardless of these reasons (Fraser *et al*., 2013). This criterion takes a value of (1) if a
household stated one or more reasons that may push them to evacuate and (0) if a household was not
willing to evacuate;
• C2: the availability of a self-host destination (Chang et al., 2009). This criterion was coded (1) if a
household had one or more relocation place(s) and (0) otherwise;
• C3: the capacity to move from the area by their own means of transport (Luathep *et al*., 2013). A
value of (1) was assigned if respondents stated that they would leave their place of residence by private
car and (0) if they stated they would use other means (public transport, close relative's car, means of
transport provided by public authorities or thanks to solidarity, etc.) or did not know;
• C4: access to the workplace or possibility of working from their evacuation destination, as work
obligations could reduce the likelihood of evacuation (Mesa-Arango *et al*., 2013). Respondents who



answered that they would be able to keep going to work or keep working at their relocation place were
coded (1) and (0) if they would not;
• C5: the presence of vulnerable people in the household (Lim *et al.*, 2016). This criterion took a value
of (1) for a household with no particular constraints relating to physical capacities and (0) if the
household had one or more particular condition.
These criteria were chosen because they are the most reliable ones which best reflect the tangible (and
therefore observable) factors of evacuation. They also correspond to significant factors frequently
mentioned in the literature.
The definition of the typology broken down into two levels. The first level contains 4 types:
• Type 1 (T1) => totally autonomous: all above criteria with the value "(1)";
• Type 2 (T2) => partially dependent: declared one or more reasons that could push them to evacuate
(C1=1) and at least one other criterion with the value "(0)" above;
• Type 3(T3) => totally dependent: declared one or many reasons that could push them to evacuate
(C1=1) and all other criteria with the value "(0)" above;
• Type 4 (T4) => not willing to evacuate: declared that they were not willing to evacuate (C1=0).
The second level consists of splitting type 2 (T2) into types "2a, 2b, 2c and 2d" according to the
criteria that make the respondent partially dependent in the event of evacuation
• Type 1 (T1) => totally autonomous: all criteria above with the value "(1)";
• Type 2a (T2a) => declared one or more reasons that could push them to evacuate (C1=1) and partially
dependent with regard to the relocation place (C2=0) and/or the means of transport to get there (C3=0)
only;
• Type 2b (T2b) => declared one or more reasons that could push them to evacuate (C1=1) and partially
dependent with regard to the possibility of continuing going to work or continuing working at their
relocation place (C4=0) only;
• Type 2c (T2c) => declared one or more reasons that could push them to evacuate (C1=1) and partially
dependent with regard to constraints relating to physical capacities (C5=0) only;
• Type 2d (T2d) => declared one or more reasons that could push them to evacuate (C1=1) and partially
dependent with regard to a combination of two criteria (C2=0 and/or C3=0 and/or C4=0 and/or C5=0)
apart from the combination of "having a relocation place (C2=1) and a private means of transport to
get there (C3=1);
• Type 3(T3) => totally dependent: declared one or more reasons that could push them to evacuate
(C1=1) and all other criteria with a value of "(0)" above;
• Type 4 (T4) => not willing to evacuate: declared that they were not to be willing to evacuate (C1=0).
To simplify the explanation, the following classification tree (see fig.3) presents the combination of
criteria for each group in the second level of the typology.


The descriptive statistics are then used to describe each type. The aim is to highlight any existing
criteria common to all the types with regard to socio-demographic characteristics together with the factors
for against evacuation. Finally, the results are completed by a brief analysis of the residents' expectations
regarding the preparation of the evacuation process and the related information (cf. section 4.3).

### 3.5. Sample profile of the respondents

The sample structure shown in Table 1 reflects the highly specific character of the inhabitants of the
"Front de Seine" towers in the 15th district of Paris with a high average age (84% are over 45 years old,
48% over 65), households composed mostly of one or two people (82.6%), a small majority of retired or
inactive residents (51.5%) and respondents having lived in this neighborhood for an average of 16 years.
Few of the respondents have a pet (14%) and a majority of households own a car (51.8%), which is
explained both by a higher standard of living than the neighborhood average (according to information
collected from the building managers who know their residents very well) and by the existence of a
dedicated car park (quite rare in Paris).
The slight over-representation (48%) of people over the age of 65 in our sample (according to the
building managers) is explained by their greater availability, their interest in security issues and an
awareness of being more vulnerable or dependent on their surroundings if evacuation is necessary. Their
vulnerability is exacerbated in the event of power supply failures that would oblige them to leave the multi-
floor residential buildings without the benefit of an elevator. Moreover, other categories of people might
not only feel unconcerned, but they might also be wrongly informed about the topic. Arlikatti *et al*. (2006)
and Zhang *et al*. (2004) stated that risk-area maps do not necessarily allow some people to understand that
an evacuation warning applies to them and therefore consider that they are not particularly concerned by
the evacuation survey.
The high proportion of respondents living alone or in a couple (49% and 33% respectively) reflects
the trend in Paris as a whole and in the 15th district, where 51% of the population live alone (INSEE, 2019).
Among the respondents, 48% are over 65, and 4% have reduced mobility – characteristics that must
be taken into account in the event of an evacuation without elevator. This vulnerable population is clearly
identified by the building managers as they know they have to prioritize them. This raises the question of
coordinating the evacuation of the different categories of people in the building by the building manager(s).
It also raises the question of their training, in so far as they claim that they have not received specific
instructions regarding this type of situation.






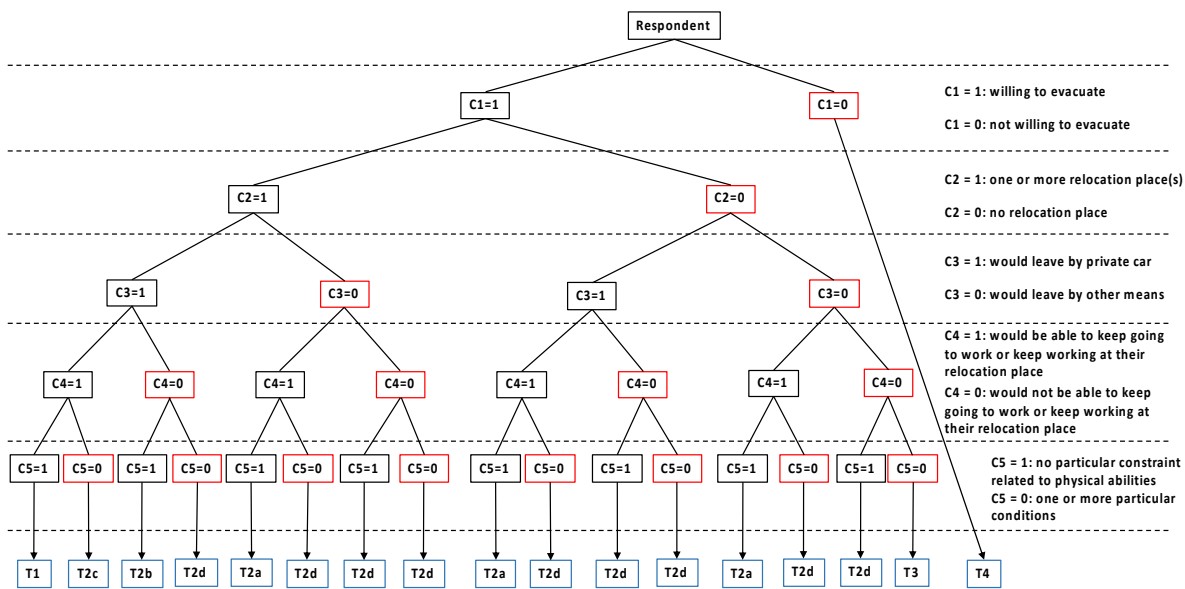


**Fig. 3.** Household typology according to evacuation capacities (second level of typology)





**Table 1.** Respondent's characteristics

| Variable | Sample |
|---|---|
| ***Respondents' demographics*** | |
| **Gender** | % (n= 522 ) |
| | Female     / Male |
| | 57.1% (298) / 42.9% (224) |
| **Age group** | % (n = 517) |
| under 25 | 0.9 (5) |
| 25 to 45 | 15 (78) |
| 45 to 65 | 35.8 (185) |
| Over 65 | 48 (249) |
| **Number of people in the household** | Study area <br> % (n=512) |
| 1 | 49.4 (253) |
| 2 | 33.2 (170) |
| 3 | 9 (46) |
| 4 | 6.8 (35) |
| 5 or more | 1.5 (8) |
| 3 or more (total 3-4-5) | 17.3 (89) |
| **Occupation** | % (n=520) |
| Active | 48.4 (252) |
| Retired | 45.2 (235) |
| Inactive | 1.7 (9) |
| Active and retired | 4.6 (24) |
| ***Other characteristics*** | |
| **Floor** | % (n=514) |
| 0 to 8 | 17.3 (89) |
| 9 to 16 | 34.6 (178) |
| 17 to 24 | 26.7 (137) |
| 25 to 33 | 21.4 (110) |
| **Year of installation** | **% (n=510)** |
| 1970-1980 | 17.4 (89) |
| 1981-1990 | 15.5 (79) |
| 1991-2000 | 17 (87) |
| 2001-2010 | 20 (102) |
| 2011-2019 | 30 (153) |
| **Own an animal** | % (n=523) |
| No | 87.1 (456) |
| Yes | 12.81 (67) |


| Own a car | % (n=523) |
|-----------|-----------|
| Yes | 51.8 (271) |
| No | 48.2 (252) |


## 4. Results and discussion

### 4.1. The main constraints on the respondents

Globally speaking, the majority of residents are not subject to tangible constraints in the event of evacuation. A little over half the households in our sample (52%) own a car and could be autonomous during an evacuation. Some 32% declared that they counted on the public authorities to provide them with a relocation place and 7% stated that they did not know where to go. This will be discussed below. Generally speaking, the households own no pets, but those who own at least one (13%) seem to be attached to it. When asked about any particularities of the household to be taken into account in the event of evacuation, some specify that they have a pet and indicate the number of pets living there. This type of person might not be willing to evacuate.

The analysis of responses in terms of expectations and information needs in the event of the need for evacuation reveals high expectations in terms of support from the public authorities.

Most residents seem to have a correct perception of the flood risk and evacuation procedures in their area, or at least to be aware of the issue. Only 15% think that their area has never been flooded. As mentioned above, a huge part of the Parisian territory, including a major part of the 15th district, was completely flooded in 1910. Some 64% of respondents know that their area might still be flooded despite all the infrastructures built to control rising waters. This result shows that residents are well aware of the limitations of the structural measures. This can be seen as evidence of progress in flood risk awareness led by the Seine-Normandy basin stakeholders. On the other hand, they have distorted ideas relating to specific but essential technical points. This affects their perception of the magnitude of the consequences of a major flood, which would necessitate preventive cuts of urban technical networks. Some 54% think that their building has a generator that will guarantee their electricity supply for at least 4-5 days. However, the generators have only 24 to 48 hours' autonomy and while they are present in every building, most of them are located underground and are therefore vulnerable to groundwater.

The last important result relating to the level of knowledge about evacuations is that 46% of the respondents are aware that the public authorities cannot host all residents of the high-rise buildings. Some 45% declared that they did not know whether the public authorities have this capacity or not. This could be linked to a statement made by one respondent, essentially claiming that, "*The public authorities objectively might have the means to host everyone but it might not be their priority, or they might have their own reason not to be willing to do so*". Debating whether the public authorities should indeed host everyone falls outside the scope of this study. It actually raises a much broader and hotly debated issue of public policies and the sharing of responsibilities in such a situation (Godfrin *et al*., 2002). In order to


provide analyses that can used more directly, we prefer to acknowledge the existence of law n° 2004-811
on the modernization of civil security. It would therefore be more relevant to identify the conditions in
which the evacuation process could be efficient.

People's perceptions vary considerably as far as this law is concerned. According to the present study

results, 52% agree while 39% disagree and the remaining 9% have no opinion on the matter. However,
such perceptions do not systematically reflect the same meaning. People subject to no constraints, for
instance, sometimes disagree with this law not because of their own situation but for the sake of vulnerable
individuals who need assistance. Nonetheless, such a perception might not exactly reflect their actual
opinion. In reality, when answering the question, people might have thought that this law applies to persons
with reduced mobility as well, but this is not the case. The results (people's opinions) would ideally require
further explanation, especially in the case of those who declared that they disagree with law n° 2004-811.
In the end, this global trend in the level of knowledge about the flood risk and evacuation procedures is
rather reassuring because one of our hypotheses was that the residents have mistaken perceptions about
the flood risk. In light of these global perception trends, many respondents have what would appear to be
the correct perception of the risk and the evacuation conditions.

As for the evacuation process, 60% of the respondents expect to receive evacuation advice from the

public officials between 24 and 48 hours before the water reaches their area. This means that a lot of people
count on the capacity of the public authorities to anticipate the event, whereas the matter is actually more
complex than that. In fact, at the end of the survey, some respondents specified that evacuation should be
recommended only if this is genuinely necessary. The problem here is that there is no guarantee that
advising residents to evacuate 24 to 48 hours beforehand would be relevant. Naturally, anyone involved is
faced with uncertainty whenever they are in a context of natural hazards. More precisely, the predicted
flooding and evacuation scenarios can never be a hundred percent reliable. The public authorities often
forget to take this element of uncertainty into account in the crisis management process. The contribution
of Kolen (2013) is important in light of the need to implement effective safety strategies despite the
uncertain nature of flood risks.

The perception of the timing during an evacuation process might help in anticipating people's

behavior. Among those who own a car, 43% declared that if they received an evacuation notification, they
would wait at home and see how critical the situation got. A further 28% would leave home within 24
hours and only 12% would leave immediately. Most people would therefore remain at home and judge for
themselves if they need to leave. The problem ascertained by Alou (2018) is that people sometimes have
difficulty in obtaining the right information about a situation that would directly affect them, thereby
causing them to evacuate too late. This statement is accurate in the case of high-rise buildings residents.
The information gleaned from the media affects them differently in comparison to residents of smaller
buildings. The point at which their electrical generator is flooded might be different from the time other
buildings are flooded at some level (underground or not). This means that they have to be informed more
directly via the building managers and the managers of the underground parts.



The survey probed the Parisians on the reasons which would decide them to leave their tower for
several weeks in a situation of major flood of the Seine (see question 11 on the appendix). Among the 10
reasons proposed, three main reasons to evacuate were reported by the residents: evacuation advice from
the public authorities (71%), the degradation of everyday commodities inside and outside their home (52%)
and the existence of a private or a public relocation place (50%). The first reason reflects the same findings
as those obtained by Baker (1991), Dash & Gladwin (2007) and Kreibich *et al.* (2017): official warnings
are important factors of evacuation decisions. Of course, this is underpinned by a certain number of
conditions, notably the communication channel used and the clarity of the message, as reported by Baker
(1991), Paul & Dutt (2010), Parker (2017) and Gissing *et al.* (2019). The two other main reasons (i.e.
degradation of everyday commodities inside and outside their home and the existence of a private or a
public relocation place) have a greater direct impact on people than other reasons mentioned in the
questionnaire such as seeing the neighbors leave, information in the media, etc. As is commonly found,
expected personal impacts strongly incite people to protect themselves and better anticipate an evacuation
(Fritzpatrick & Mileti, 1991; Huang *et al.*, 2012; Lindell & Perry, 1992).
To go further in the analysis, an ascending hierarchical classification performed on the ten evacuation
reasons (variables) with the Sphinx iQ2 software (fig.4.a and fig.4.b). It highlights the groups of
explanatory reasons for the propensity to evacuate according to households profiles.

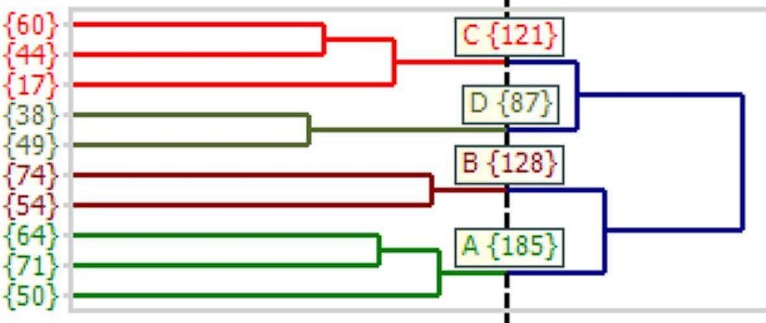


Fig.4.a. Dendrogram of the question 11 (in appendix) with 521 complete observations on a total of

523.

| | |
|---|---|
| A (185) | + q11i, q11g<br>- q11e, q11d, q11j, q11c, q11a |
| B (128) | + q11e, q11b, q11f, q11h<br>- q11j, q11d, q11a |
| C (121) | + q11j, q11a<br>- q11h, q11g, q11i, q11f, q11c, q11d |
| D (87) | + q11c, q11d<br>- q11b, q11i, q11g |

Fig.4.b. Characterization of classes of respondents according to 10 evacuation reasons (variables q11a
to q11j).



The dendrogram in fig.4.a allows to identify four groups of respondents according to the classification
of answers group they gave. The characterization of classes of respondents (fig.4.b) shows for the variables
in green, the mean values of the class are significantly higher than those of the rest of the sample. The two
main decisive reasons for evacuating are knowing that your accommodation is in a secure area and having
a private or a public relocation place (group A: 185 respondents on fig.4.a). The analysis confirms too that
people are awaiting public or mediatic and precise information and information on the consequences of a
refusal to evacuate before taking their decision (group B, fig.4.b).

*4.2. Typology of households according to evacuation capacities*
The first level of typology, which distinguishes autonomous households from others, shows that most
respondents (77%) are partially dependent in the event of evacuation (fig.5). We named this group T2 on
fig.2. This initial information is not surprising. It leads to further analyses in order to better understand the
factors that make this group partially dependent and to anticipate the actions to be taken in order to
guarantee security when evacuating. That is the object of the second level of typology, explained below
(fig.3). Among those people who are totally dependent (group T3, accounting for 14%), there are many
old people who may be somewhat socially isolated. They may have neither a relocation place nor a private
means of transport to get there. These old people are automatically classified in group T3 as they display
all the criteria of a lack of autonomy. As for the few respondents in the group T4 who declared that they
would not to be willing to evacuate, such a statement has to be taken with some caution. It is to be included
in the typology, although it is not a directly observable variable because it is a crucial information.
Nevertheless, a number of building managers stated that when they attempted to initiate an evacuation
exercise, people were definitely not reactive. The reasons for this could not be formally verified, but it may
mean that the residents are not convinced of the necessity for such an exercise. If so, they might also not
be convinced that one day they could actually be asked to evacuate. This small proportion of T4 could
therefore be misleading. In a real context of flooding and evacuation advice, the different actors involved
expect that a larger proportion of people would not be willing to evacuate. Further explanations for this
will be provided later in this paper.
The second level of the typology splits T2 (partially dependent) into T2a, T2b, T2c, and T2d (fig.3).
Fig.6 reveals that many people are partially dependent, mainly because they do not have a relocation place
and/or a private means of transport to use (T2a accounting for 55%). Hence, the issue of a relocation place
and means of transport has to be seriously considered. Furthermore, the global tendencies described above
reveal that knowing where to go in the event of an evacuation is one of the three main reasons that could
incite people to evacuate. This also reflects the fact that most people may actually rely on public authorities
with regard to these two elements (relocation place and means of transport). Consequently, the public
authorities might have to anticipate a double phenomenon in the event of evacuation: (i) the first level of
typology reveals a very small number of people not willing to evacuate, but many others might also not


evacuate if they do not know where to go or how to get there; and (ii) for those who are willing to evacuate,
most of them count on the assistance of the public authorities. Even the proportion of T2b (12%) confirms
that the relocation place and mobility are key issues because people in this category are not certain to be
able to continue going to work or working at their relocation place. This break-down of T2 helps us
understand why the debate about law n° 2004-811 is so sensitive and often beset by controversy, given
that one of the critical issues is the relocation process. The analysis of access to relocation places could
therefore be refined through more formal models and more detailed qualitative interviews.

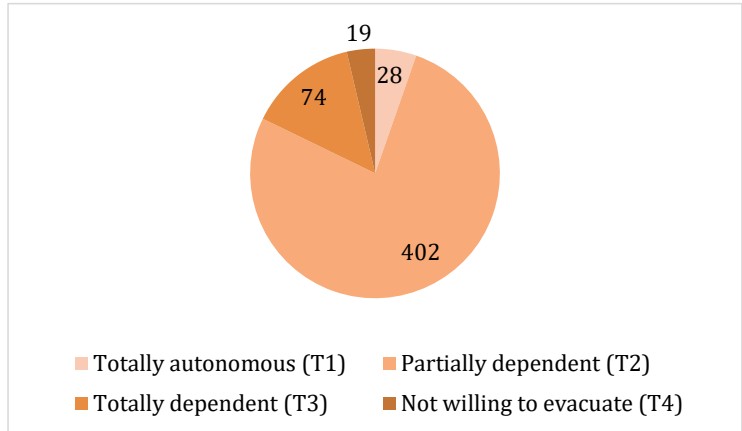


**Fig.5.** Typology with respect to the respondents' evacuation capacities (first level of typology)

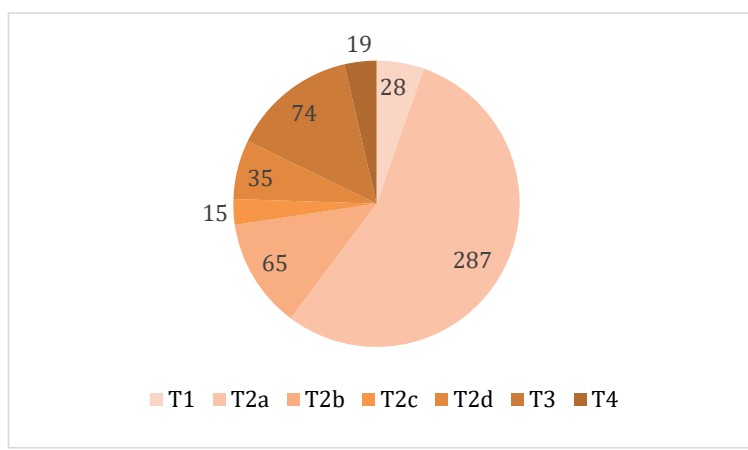


**Fig.6.** Typology with respect to the respondents' evacuation capacities with detailed types of
partially-dependent people (second level of typology)
These arguments lead to a more detailed analysis of who belongs to which type, with three main
descriptive categories:



- A comparison of the 7 types considering the socio-demographic variables of age and gender. Age inevitably needs to be analyzed because the relationship between old age, isolation and mobility has already played an important role in this study. Gender will also be analyzed here because at this stage, it may open up avenues for more interesting reflection. It was not mentioned earlier in this study because even though some authors, such as Whitehead *et al.* (2000), found that women were more likely to evacuate, our hypothesis is that gender has no effect on evacuation decisions and capacities, echoing the results of Baker (1991), Dow & Cutter (1998) and Huang *et al*. (2016);

- A comparison of the 7 types considering the perception of law n° 2004-811. This perception can be better interpreted now that we have divided the respondents into seven types. It is mostly important to understand whether certain types tend to hold the same opinion on this law. Furthermore, such a comparison would help distinguish those who are subject to physical constraints and might have stated that they disagree with this law. As explained above, such a declaration might actually be biased because self-evacuation and self-hosting, as stated in law n° 2004-811, does not apply to people with reduced mobility;

- A comparison of the 7 types considering two variables that could add significantly more capacities or constraints to the evacuation process, namely possession of a vehicle and the level of the floor where the respondent lives.

With respect to type and age group, the distribution shows that a large majority (59%) of the individuals totally autonomous (category T1) are aged between 45 and 65, and 30% are over 65. For those who are partially dependent regarding the relocation place and/or the means of transport to get there (T2a), the proportions are quite similar between the 45-65 group (43%) and the over-65s (39%). Moreover, the older the residents are, the less likely they are to be able to continue going to work or continue working at the relocation place. Among those who are totally dependent (T3), 66% are over 65 years old. In T2c (partially dependent regarding the particular constraint related to physical abilities), half are relatively young, aged between 25 and 45. This is normal because the older residents would display the numerous criteria underpinning a lack of autonomy, which is why they would belong to categories other than T2c. These results show that type and age group are often linked to one another.

The classification according to gender is standard, with 55% women, 40% men and 5% indicating both genders because they might have completed the questionnaire together. Women are predominant in T2a (60%), T3 totally dependent (63%) and T4 not willing to evacuate (58%). In contrast to our hypothesis, they might therefore be more vulnerable than men. Incidentally, while they might be more vulnerable, they are not more likely to evacuate, again in contrast to our hypothesis. In such a modern society, it is difficult to provide any explanation for such a trend. Rather than reusing these results, it would better to conduct a new survey or interviews to control for different possible factors of a socio-psychological, physical or other nature.

The result of classification with respect to type and opinions concerning law n° 2004-811 on the modernization of civil security is very coherent. Respondents displaying negative opinions (38% in total),



meaning that they do not approve the law, are clearly predominant in group T3 (totally dependent, 40%) and T4 (not willing to evacuate, 42%). On the other hand, those who agree with the law are predominant in all other types. In T2a, there is very little different between the proportion of those who agree with the law and the share of those who do not. Once again, this reflects the different situations of the residents, as far as evacuation is concerned, who do not have the same opinion about the law within their own group. This opinion should be clarified in further studies.

Furthermore, when people do not own a vehicle (48% in total), they mostly whether belong to T2a (65%) or T3 (totally dependent, 69%). Again, such proportions are coherent. As the proportions of those who do not own a vehicle in these two types are significant, this distribution effect gives the impression that only those who own a vehicle belong to the five other types, which does not necessarily make sense. Incidentally, 93% of those who own a vehicle belong to T1 (totally autonomous). However, owing a vehicle does not guarantee total autonomy. Independent of owing a vehicle, autonomy also depends on the priority criteria defined in our methodology (fig.3).

Last, the level of the floor is quite random for most types except, in two cases. In T1, 46% live above the 24th floor, which means that the most autonomous people tend to choose to live on the upper floors. On the contrary, 16 of the 19 people in T4 (not willing to evacuate) live below the 17th floor. They probably focused on the issue of the elevator, thinking that it would not affect them if it stopped working because they felt able to cope on their own. This data could prove useful in improving information for residents in the event of evacuation and to dispel misconceptions.

### *4.3. Respondents' expectations regarding evacuation information and preparedness*

### 4.3.1. Information as a priority issue

Here we present a brief analysis of the residents' expectations regarding preparation of the evacuation process and the associated information. To this end, a word tree was generated from the text contained in the 521 responses to the open-ended question 17: "what would you like to be done so that you would be better prepared in case you need to leave?" (see questionnaire in Appendix) (fig.7).

This text is transformed into a visual tool where the words are arranged in a tree-like branching structure which reveal recurrent ones and indicates the strength of their semantic proximity in the text. The word tree visualization method consists of counting the frequencies or repetitions of quoted words for calculating their semantic proximity (Wattenberg & Viégas, 2008). For this, we used the open source online application "www.treecloud.org" (where the algorithms were implemented by Gambette & Véronis, 2010). The figure which one obtains consists of branches of words or "edges". These edges are all the longer as the word classes are the most significant (close to each other, well separated from the rest on the figure). This visualization tip improves readability compared to a simple word cloud. The advantage of the tree view is also to benefit from a better amount of information (represented by a number of groups or "bags" linear nested words) and better quality of information (considering global information by matching

words in the tree). The coloring of the words guides the reading according to different possible criteria
(their frequency of use in the responses, their chronology in a speech, etc.).

Here in the fig.7, the font coloring associated with the words is linked to their frequency (from light
blue for the little cited word to red and bold for those cited several times). When comparing the branches
of the tree built from the most frequent words used in the respondents' opinions gathered from question
17, these following conclusions arise. The respondents most often cite the word "information", which
appears in red in the longer branch of the tree, upper right on the figure. In this branch of words, the word
"information" is associated in descending order with the word "evacuation", then "instruction" and
"know". In the symmetric branch (on the bottom left of the figure), the words "informed", "case",
"advance" are among the five words which have the highest frequency; in addition to "flood" and "should".
Thus, the idea of being well informed, especially on the practical modalities on "evacuation", is the priority
for the respondents who live in the Seine front towers.
In fact, people very frequently ask to be informed about numerous details regarding the evacuation
process. Instead, they could have requested some form of help, for instance, but very few people thought
of it. Together with information, people wish to receive clear instructions in good time so they can prepare.
Some mentioned that receiving instructions at an early juncture would help them prepare their relocation
place. As Dash & Gladwin (2007) explained, "warning is an integral component of evacuation decision
making". Others replied that they will follow the information provided by the authorities. This echoes our
previous finding relating to the importance people give to instructions and evacuation advice from the
public authorities. Some respondents also pointed out the need for an evacuation drill, with some of them
who even specified the expected frequency of such a drill; for example, once or twice a year. The question
of communication is also addressed by the respondents through the recurrence of the words
"communication" and "meetings". They would like to have regular meetings about the situation and to be
given pamphlets presenting the risks and safety measures. In reality, people might not use these means of
communication (pamphlets, Internet and others), but sharing them might improve peoples' knowledge and
consciousness, if only to a small degree.


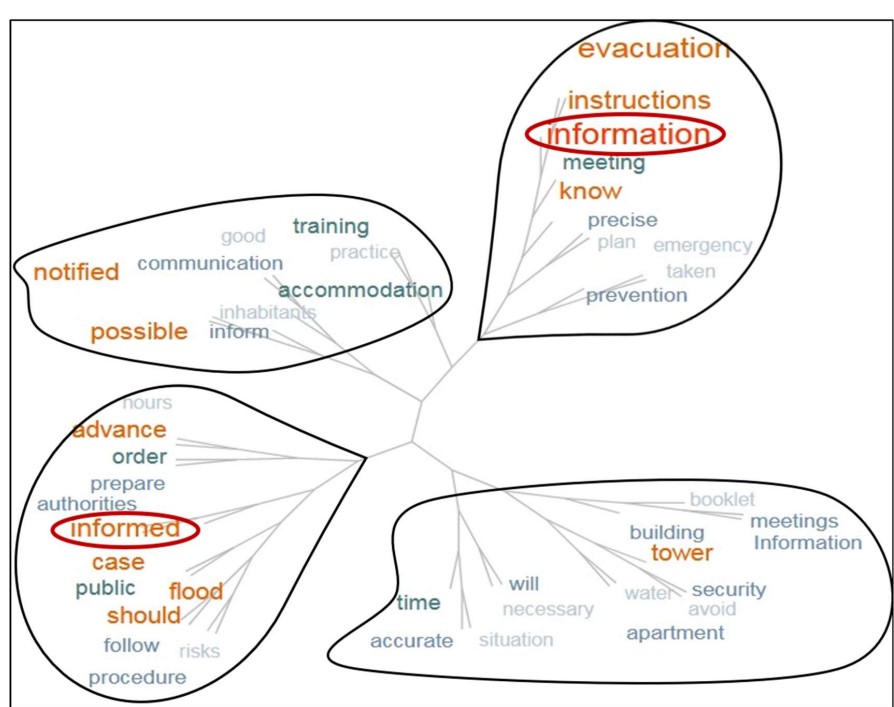


**Fig. 7.** Word tree of the respondents' expectations in order to be better prepared for an evacuation
**4.3.2. Implications on information dissemination practices**
The importance of information is clearly described by Colbeau-Justin & de Vanssay (2001) through
their case study conducted in the *département* of Somme in France. Due to the lack of information and
formal and sustainable information channels both before and after the flooding, there were rumors about
and denial of the flood risk. Becerra *et al*. (2013) mention examples where such a phenomenon led the
authorities to introduce alarm systems. Such an experience shows that information is crucial and because
it is requested by the residents themselves, it is a form of responsibility that they assume, as it helps in
preparing themselves for a "crisis".
In our case study, rumors about and denial of the flood risk are not the only issues as far as the
knowledge of the people is concerned. In fact, the textual answers reflect a very approximate knowledge
of the person responsible for one or other action – for example: who sets the alarm? Some think that the
prefecture has to deal with all tasks related to evacuation. Generally, the distribution of the public officers'
functions is clearly explained on internet. People therefore need to be better informed through more diverse
means (including flyers). This erroneous information could be due to the fact that those people have never
experienced the situation at first hand and have never paid attention to such a detail (though it cannot really
be called a detail). Another possible cause is the increasing complexity of the actors' systems (Becerra &


Peltier, 2011). This is particularly true in the case of crisis management not only in Paris as a metropolis,
including in the context of a flooding, but also in France in general.

In response to this lack of knowledge, Becerra *et al*. (2013) suggest "personalizing the risk". This idea

has already been mooted by Thouret & D'Ercole (1996), who established that repeated personalized
information which, moreover, is confirmed by many different formal sources, is necessary before the event
happens. What information, however, can be personalized in tangible terms? Much information on the
flood risk in the 15th district is already shared through meetings as well as in printed media and on Internet
(https://episeine.fr/, http://www.leparisien.fr/paris-75/83-300-habitants-du-xve-seraient-touches-par-une-
crue-centennale-04-12-2016-6412278.php). The majority of this information is therefore already
accessible. However, residents are not particularly well informed about the consequences in terms of the
disruption to services inside their building. Anyway, the person who determines and shares such
information should not create panic among the population while informing them about flood risk.

Another way to keep people informed is to encourage "intermediate actors" (Filâtre *et al*., 2005) who

would willingly receive, transfer and translate information in real time among different categories of actor
(Becerra *et al*., 2013). In the case of high-rise buildings, there are several possible intermediaries including
the building manager, the "president of the tower", or maybe a totally different person if needed. Anyway,
when providing written answers, some residents already asked for the building manager to be appointed
as the intermediate actor. This helps reinforce social participation and civic responsibility in flood
prevention (Becerra *et al*., 2013).
*4.4. Limitations and perspectives of a first-step study in a particular context*

Ultimately, it should be recalled that in such a prospective study, there is always a gap between

perceptions and behavior in a real context of flooding. Although the results revealed that only a few people
would not evacuate, other people's opinions should not be self-sufficient. It is certain that the better
informed people are (notably with a clear, more specific warning), the more they react accordingly (Mileti
& Beck, 1975). However, even being well informed does not entirely guarantee that the real action would
be the same as the one mentioned in the completed questionnaire. Nevertheless, the descriptive statistics
showed some particularly coherent answers, for example for T1 (totally autonomous), T2a (partially
dependent regarding the relocation place and/or the means of transport to get there) or T3 (totally
dependent).

Across all the results and analyses, one main limitation was observed: the survey was not sufficiently

detailed to provide all relevant explanations. There is therefore a need for further analyses of the different
factors which explain the perceptions of and reasons for evacuation such as personal experiences,
knowledge and characteristics to name but a few. Moreover, the survey did not directly examine the
reasons why people would not evacuate, according to their own perceptions. This could help in anticipating
evacuation behavior. This idea of explaining the reasons not to evacuate is inspired by the works of other
authors such as Baker (1991), Dow & Cutter (2000), Riad *et al*. (2006) and Kolen (2013).



Furthermore, this study could not explore all the particularities of the case of high-rise buildings. One
such particularity is that living in a high-rise building could provide a certain feeling of security. This idea
was implicitly evoked throughout our analyses but could not be formally confirmed as there were no direct
questions on this matter. In fact, the perceptions of people living in smaller buildings differ from that.
Many authors found that residents feel much more concerned when they are convinced that there is a risk
of serious injury to themselves, their families or of damage to their homes (Baker, 1991; Gladwin *et al*.
2001; Huang *et al*., 2012; Riad *et al*., 2006; Lindell *et al*., 2005; Whitehead *et al*., 2000). This means that
when faced with the same hazard, in the 15[th] district of Paris for example, the residents of high-rise
buildings and those of small buildings would not take the same decision concerning evacuation.
Finally, this paper highlighted a certain number of results that could inspire broader studies in
geographical terms. This could be the level of knowledge in the event of evacuation (for example who
does what or what the flood risk is in the area concerned? etc.) or the opinion on law n° 2004-811 (in a
much larger survey, would opinions still be as mixed as they are in our case study? Why?). Even the
proportion of people willing to evacuate or not and their evacuation capacities vary geographically. All
these issues can be explored through further studies.
**5. Conclusion**
This paper addresses evacuation issues in the case of the Parisian metropolis following major flooding
with slow kinetics. The central question concerns the proportion of people who are willing to evacuate, the
constraints they face and their capacity to self-evacuate, self-host and reach a relocation place. The overall
approach relies on a prospective study based on a survey conducted in a Parisian area on the banks of the
River Seine, and more particularly in high-rise buildings.
The main typology results, those of a, revealed that the majority of the respondents would be partially
dependent in the event of an evacuation. More precisely, one group among them is predominant: those
who do not have a relocation place and/or private means of transport to get there. Ultimately, after
comparing all the detailed results, the relocation process is the main issue of concern to the residents,
especially the older ones. In total, four factors are shown to be important to people and could encourage
them to evacuate: (1) the evacuation advice from the public authorities, (2) the fact that they know they
have a relocation place and can get there, (3) the disruption of the facilities in their building, and (4) formal
and clear information about the hazard and its consequences. The different actors have to better anticipate
the evacuation behavior by taking these factors into account.
Furthermore, the matter of approval of law n° 2004-811 on the modernization of civil security was
addressed in this paper. Our study provided certain explanations underpinning the reasons why this law is
controversial. One possible way to make it more efficient is to run general and personalized information
campaigns on the risk of flooding, its consequences and the adaptive reactions. The literature also
emphasizes the aspect of risk perception. This study helped provide a global view of the trend in
perceptions, but it is limited regarding explanations.



Anyway, this paper proposes another perspective in the field of flood risk and evacuation surveys: it
is a study dealing with anticipation, while most studies focus on past experiences. In fact, the public
authorities do not, at present, have information on people's capacity to self-evacuate, reach a relocation
place or self-host. Are the residents of high-rise buildings prepared for evacuation? They are not that well
prepared and this study provides details relating to this without waiting for a disaster to occur in order to
learn from it. Another major contribution of this paper is the perspectives it offers on preparation for
flooding, in particular with slow kinetics. This raises specific issues relating to information and the
coordination of an evacuation as the actors and populations normally have time to prepare themselves for
the crisis. Moreover, people might be dimly aware of the consequences of progressive flooding, which
does not give rise to emergency evacuations. Finally, this study is a first step towards a possible broader
geographical analysis of people's perceptions and capacities in order to better prepare themselves and the
authorities for evacuation in moderate risk areas. To deepen this prospective research, the team of the
RGC4 project also conducted a survey in ex-post situation in the suburbs of Paris that were flooded and
affected during the 2016 and 2018 Seine floods and its tributaries. It will be particularly interesting to
compare the results of these two recent surveys. Furthermore, other methods could complete this step,
notably modelling. This might consist of predicting the proportion of people willing to evacuate and the
timing of evacuation, a very essential estimate for decision support.


**Appendix – Questionnaire sent to the residential high rise building households near the Seine**

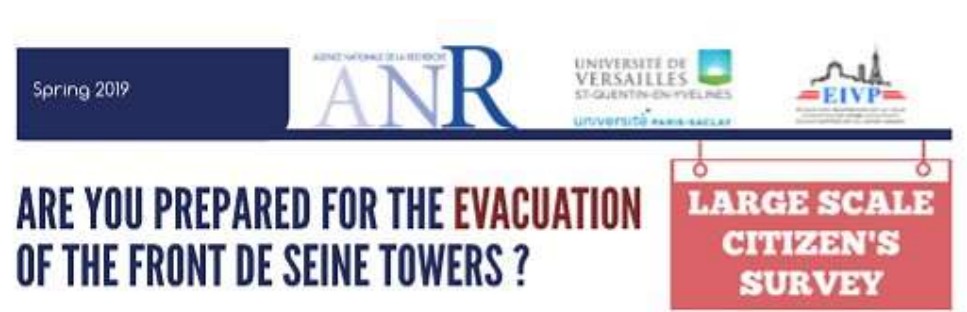

Spring 2019   ANR   UNIVERSITÉ DE VERSAILLES ST-QUENTIN-EN-YVELINES   université PARIS-SACLAY   EIVP

# ARE YOU PREPARED FOR THE EVACUATION OF THE FRONT DE SEINE TOWERS ?

**LARGE SCALE CITIZEN'S SURVEY**

WHY? Because evacuation will be mandatory in case of a long time blackout subsequent, for instance, to an exceptional flooding of the Seine. The blackout will put the elevators out of service, and in your flat, everything that requires electricity will also stop working!

IT IS IMPORTANT THAT YOU FILL THIS SURVEY. It will be useful in providing information on your ability to leave your tower and join a safe place.

WE EXPECT YOUR RESPONSES in order to make recommendations to the municipal services, the emergency and crisis management services. The objective is to better inform and accompany you in such a situation.

THANK YOU FOR PARTICIPATING.

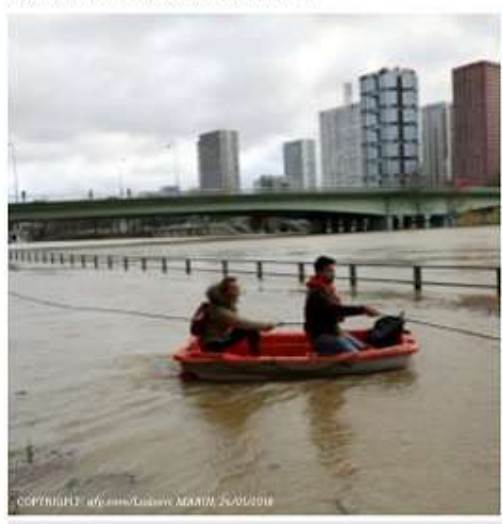

COPYRIGHT: afp.com/Ludovic MARIN 26/05/2018

**NAME OF THE PROJECT : "RGC4"**

Urban Resilience and Crisis Management in a context of Slow Kinetic Flood in Grand Paris, project lead : Engineers' School of the City of Paris, 80 rue Rebeval 75019 Paris. (https://urlz.fr/9Eig)

**FUNDING :**

National Research Agency (https://anr.fr/Project-ANR-15-CE39-0015)

**PARTNER in charge of the survey and contact :**

Mme Nathalie Pottier, Teacher-Researcher nathalie.pottier@uvsq.fr

CEMOTEV Laboratory of the University of Versailles St-Quentin-en-Yvelines, 47 Bd Vauban 78047 Guyancourt

The Municipality of the 15th district, the City of Paris and the Paris Prefecture are aware of this survey.

Your area in the 15th district gathers the most numerous and tallest buildings in Paris, by the Seine riverside. This is why we have chosen it as our pilot survey with 14 towers.

The floodings in 2016 and 2018 in the Parisian region showed that the disturbances extended beyond the flooded area (transportation, degradation of the basic services).

Let us get prepared altogether.

THE REPONSES COLLECTED WILL BE ANONYMOUS

Are you interested in the results?

A synthesis of the results will be shared to the residents in autumn 2019.

You can also express freely your opinions about the subject on a paper that you will attach to this questionnaire.

**HOW TO GIVE THIS QUESTIONNAIRE BACK?**

Thank you for putting this questionnaire in a closed envelop and dropping it into the drop box on the reception desk of the tower.

Thank you for replying AS SOON AS POSSIBLE, by June 15th, 2019 (in case you were away, we can accept belated filled questionnaire that you will leave at the reception desk or by mailing to the partner's address but the sooner, the better!)




1. What is the name of your tower? _______________________________

2. Which floor is your flat on? _______________________

3. When did you move in this tower (date or year)? _______________________

4. Have you got any pet(s)?

○ No

○ Yes, and they are easy to transport in case of evacuation

○ Yes, but they are hard to transport or are cumbersome in case of evacuation

○ Other. Specify whether they need special precautions in case of transportation (animal in a cage or dangerous...):

_______________________________

5. Do you know what to do in case of an evacuation advisory?

○ Yes          ○ Partly          ○ No

6. If you have got a vehicle, on which level is it parked?

○ Parking -2 under slab     ○ Parking -1 under slab     ○ Underground parking elsewhere     ○ On surface

7. If you receive an advisory to evacuate the underground parkings due to a flood, WITHOUT any other advisory to evacuate the towers, what will you do?

| | I move my car away and... : | I haven't got a car |
|---|---|---|
| I stay home and better assess how hazardous the situation is | ☐ | ☐ |
| I prepare myself to leave home within 24h | ☐ | ☐ |
| I take this opportunity to leave immediately | ☐ | ☐ |
| I do not know what decision I would make | ☐ | ☐ |

8. What do you think of the following statements?

| | TRUE | FALSE | DO NOT KNOW |
|---|---|---|---|
| This area has never been flooded | ☐ | ☐ | ☐ |
| An exceptional flooding of the Seine in Paris is predictable at least one week before | ☐ | ☐ | ☐ |
| Thanks to all of the infrastructures (dams, murettes, etc.), this area cannot get flooded at all | ☐ | ☐ | ☐ |
| The tower has power generators that guarantee electricity autonomy for at least 4 or 5 days | ☐ | ☐ | ☐ |
| In case of blackout in the tower, I can conserve tap water and waste water evacuation system | ☐ | ☐ | ☐ |
| The public authorities are able to host and/or rehouse all of the residents of the towers | ☐ | ☐ | ☐ |



**9. Except for fragile people, the modernisation law of the Civil Security recommends the self-evacuation and self-hosting BUT NOT JUST EXPECTING the help from public authorities. Do you agree?**

○ I totally agree         ○ I totally disagree
○ I would rather agree    ○ No opinion
○ I do not really agree   Specify your opinion: _______________________

**10. If the prefecture issue an evacuation advisory linked with a major flood of the Seine, do you think they might ask you to evacuate:**

○ Long before water invades your area (from 24 to 48h)
○ Only when the water has reached the cellar and/or the streets in this area
○ Only if the flood in this area lasts too long (several days)

**11. If you had to leave this tower for several weeks due to a major flooding of the Seine, what would incite you to make that decision? (Many possible options)**

○ **a)** Nothing, I will not leave my home in any case
○ **b)** The preventive evacuation advisories from the public authorities and the emergency services (24-48h before this area gets flooded)
○ **c)** The departure of at least half of my neighbours in my building
○ **d)** The departure of at least half of next-door neighbours
○ **e)** The information from the media or my surroundings about the degradation of the situation
○ **f)** Knowing that if I do not leave on time, I could not count anymore on the emergency services afterwards
○ **g)** Knowing that my appartment will be in a secure area
○ **h)** The deterioration of living conditions (at my place and/or in this area)
○ **i)** Knowing where I could be hosted and being able to join that place
○ **j)** Other reason (to be sepecified) : _______________________

**12. Among these reasons above (from b to j), which are the 3 first reasons which would incite you to leave?**

1st: [____]         2nd: [____]         3rd : [____]

**13. If you are given 24 to 48h to organise your evacuation before the disturbance of the transportations, where would you go? (this detail is critical for the mobility plan outside the disaster area)**

○ I have got one or many possible places to go      ○ I have no place to go, it will depend on the housing provided by the authorities

  Specify your eventual host city(/ies) or place(s) :
  _______________________                            ○ I do not know if I would leave within such a time limit
  _______________________

**14. How would you leave?**

○ By public transport                                 ○ I count on the means of transport from the public authorities, the solidarity...
○ By car                                              ○ I do not know
○ My relatives or friends, who will host me, would probably come and take me by car      ○ Other (to be specified) : _______________________




**15. As regards to working people, do you think you would be able to continue working from where you would be hosted?**

○ Yes, if means of transport are available          ○ No
○ Yes, by teleworking                               ○ I do not know

**16. Put these inconveniences in order which would make you leave if they lasted more than 3 days:**

| 1st : | 2nd : | 3rd : | 4th : | 5th : | 6th : | 7th: |

a) No more elevator                          e) No more help from the emergency services
b) No more drinking water                    f) No more public transports
c) No more toilets (backup of wastewater)    g) Other (to be specified): ___________
d) No more food supplies

**17. What would you wish to be done so that you will be better prepared in case you need to leave?**

**18. What pieces of information would be useful to you in order to leave on time in case of a generalised flooding?**

## And last

**19. The respondent is:**

○ A woman          ○ A man

**20. The respondent's age range :**

○ Less than 25 years old          ○ Between 45 and 65 years old
○ Between 25 and 45 years old     ○ More than 65 years old

**21. Number of people in your home:** ________________

**22. Particularity of people in your home to be taken into account in case of evacuation:**

○ None                                      ○ Less than 15 year old child(ren)
○ Disabled people                           ○ Elderly people
○ At least one person in your home needs a  ○ Another particularity: ________________
  regular medical or external assistance

**23. Status of the respondent:**

○ Active                  ○ Inactive (domestic work , looking for
○ Retired                   a job)

YOU CAN EXPRESS FREELY ON ANOTHER PIECE OF PAPER YOUR OPINIONS, NEEDS OR IDEAS ABOUT
"EVACUATION-FLOODING-INFORMATION-HOSTING" OR ABOUT THE SURVEY METHOD.                THANK YOU

The date will be strictly archived in an anonymous way in a computer in view to a statistical treatment and synthesis by the RGC4 project researchers. You will get the access
to that synthesis (an information about the dissemination methods will be displayed through billboards in the halls).





*Authors contributions*. This work was carried out by NR as part of her post-doctorate under the direction
of NP. She disseminated and collected the survey data with the support of NP and the help of AMES on
more technical points. The first writing and methodology for processing survey data were done by NR.
AMES, MV and NP contributed to the concept and writing, and helped with revisions as well as
proofreading.
*Competing interests*. The authors declare that they have no conflict of interest.
*Acknowledgements*. This work was developed within the framework of the research project "RGC4:
"Urban resilience and crisis management in a slow kinetics flood context. Development of tools to help
manage critical technical networks: application to Grand Paris", of the French National Research Agency.
We would like to express our sincere thanks to the public partners who facilitate the survey (the
Municipality of the 15th district, the City of Paris and the Paris Prefecture), to all our contacts like the
presidents of co-owners associations,  co-ownership managers, tenant associations (especially the
association Keller village, her president and her assistant), and the high rise buildings managers, without
whom the investigation would not have been possible, and finally, of course, households residents who
responded to our survey.
*Financial support*: This research has been supported and funded by the French National Research Agency
(ANR15 CE39 0015).

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
