# Peer review of "1. Introduction"

_Natural Hazards and Earth System Sciences, 2020_

## Referee Comment (RC1) · Anonymous Referee #1 · 6 Jun 2020

Summary: This manuscript addresses an important and unfortunately neglected issue—flood evacuation from high-rise buildings. The literature review is good but, as noted below, can be improved. The sample size is excellent, although the response rate is low and so the sample's representativeness is uncertain. However, as the authors note, the response rate is comparable to other mail questionnaires so it is not a major flaw in the study. The cluster analysis of the respondent profiles is a noteworthy innovation. The paper's conclusions

Line Comment 115 Other factors likely to affect flood evacuation decisions are environmental cues; social cues; warning sources, channels, and messages; protective action perceptions, stakeholder perceptions, and personal characteristics (e.g., sensory and physical mobility limitations, hazard experience)—see Lindell (2018). Some of these are mentioned later in the manuscript—experience on line 135, warnings on line 138, social cues on line 147, and environmental cues on line 148. These other factors should be summarized here.

128 It is indeed possible that the effects of demographic variables on evacuation is complex. Alternatively, the variation across studies in the significance of demographic variables in predicting evacuation can be explained as little more than random fluctuations that Baker (1991) characterized as small and inconsistent among studies and Huang et al. (2016) Figure I showed have consistently small effect sizes when aggregated in a statistical meta-analysis (SMA). At the very least, the authors should acknowledge that the effect of demographic variables is controversial.

134 I re-read Dash and Gladwin (2007) and I can't find any statements that support the proposition, implicit in this statement, that experience causes risk perception which, in turn, causes evacuation. In fact, Dash and Gladwin repeatedly propose experience and risk perception as competing predictors of evacuation. Also, I can't find the term "awareness" anywhere in the Whitehead et al. (2000) article.

232 Huang et al. (2016) cited 11 studies of evacuation expectations (what the present authors call prospective surveys).

243 Lindell and Perry (2012) is not listed in the Reference section.

286 Social cues are observations of other people's behavior that influence them to evacuate. The most common social cues are observations of businesses closing and other people evacuating. Social assistance is most commonly seen in people staying with peers (friends and relatives) rather than in commercial facilities (hotels or motels), government shelters, or other locations (e.g., second homes). Lindell et al. (2019) Section 6.2 summarized the US research as indicating that an average of 62% of evacuees

stay with peers, 27% stay in commercial facilities, and 3% stay in government shelters (auditoriums or gymnasiums of schools or churches).

298 The acknowledgement in this statement that some studies—especially Baker's (1991) review and the Huang et al. (2016) meta analysis, concluded that demographic variables do not seem to be significant predictors of evacuation—needs to be reconciled with the discussion of demographic variables at line 128.

337 Huang et al. (2012) also cited a response rate of 24.6% from the Texas coastal evacuation expectations survey by Lindell et al. (2001). In addition, Huang et al. (2012) p. 294 point out that the concern about low response rates is that some groups are under-represented. However, any bias in demographic characteristics is only relevant to the degree that these characteristics are correlated with evacuation expectation and the variables that are highly correlated with it. In fact, as noted earlier, the correlations of demographic characteristics with psychological variables and evacuation expectation are generally very small even when they are statistically significant. Consequently, demographic representativeness might not be as big a problem as many authors seem to think.

349 Chang (2009) is mis-cited as Chang (2019) in the Reference section.

428 I don't understand what is meant by the occupational category "inactive". The authors should explain this term.

438 Pets in the home are indeed a likely evacuation impediment, but this obstacle can be avoided if people know that there are places where they can evacuate with their pets.

504 It seems odd that the authors' conclusion about the importance of authorities did not cite the most rigorous and comprehensive review, Huang et al. (2016), because their review shows that this is the most important predictor of evacuation.

544 The discussion in subsequent lines suggests that "responsive" (which implies a

passive lack of cooperation) would be a better word choice than "reactive" (which implies active opposition).

546 Although the relationship between how people respond to a behavioral expectations questionnaire and how they actually respond in a disaster is not perfect, it is statistically significant and positive (Kang et al., 2007). Moreover, the variables that predict behavioral expectations also predict actual response in a disaster (Huang et al., 2016).

577 Baker (1991) reported a narrative review of hurricane evacuation studies conducted prior to his article and Huang et al. (2016) reported a more powerful SMA of hurricane evacuation studies conducted between Baker's review and 2014. The results of the Whitehead et al. (2000) and Dow and Cutter (1998) studies were included in the Huang et al. (2016) SMA, so it is a logical error to cite those two studies as if they provided independent evidence. A rough analogy at the level of a single study would be to find a correlation of r = 0 between gender and evacuation but argue that there is some evidence that women tend to evacuate because a subset of the women did evacuate. Given the similar findings between findings from Baker (1991) and Huang et al. (2016), any studies reporting contrary findings are most likely to be due to random sampling fluctuations.

625 The conclusion about the relationship between floor level and evacuation expectation would be stronger if supported by a ïАč2 test.

662 As a minor point, the quote from Dash and Gladwin is true for hurricanes but not necessarily for inland floods—see the Lindell et al. (2019) report on the Uttarakhand flood.

670 The information provided in hazard awareness brochures and hazard awareness meetings is not necessarily limited to those who read the brochures or attend the meetings. Lindell et al. (2015) found that more people knew they should evacuate immediately after a severe earthquake because it could cause a tsunami than had read hazard brochures or attend hazard meetings. The likely reason is that this information was passed through informal social networks either before the earthquake or immediately after it struck.

679 The reference to the Colbeau-Justin and de Vanssay (2001) and Becerra et al. (2013) papers raises the question how prevalent were rumors about and denial of the flood risk? Rumors and denial are always present, so the question is whether these were characteristic of 1% of the population or 99% of the population.

681 It is unclear what is meant by "alarm systems". Are these warning systems?

694 I think it is fine to credit Thouret and D'Ercole (1996) for presenting the concept of risk personalization but the authors should also mention Mileti and O'Brien (1992), who presented this term earlier based on Withey's (1962, p. 106) theorizing.

702 It is unclear if the authors intend "should not create panic" to mean an injunction ("We don't want this person to create panic because it is possible for that to happen.") or an expectation ("We don't expect that this person will create panic because people won't panic in response to this type of information."). If the authors intended this statement as an injunction, they are mistaken because panic is extremely rare even during life-threatening disasters—see Lindell et al. (2006) Chapter 8. If the authors intended this statement as an expectation, they should restate it that way.

716 As noted in my comments on line 546, there is relevant research on the relationship between expected and actual evacuation behavior.

726 Asking people to endorse specific reasons why they didn't evacuate seems like a good idea, but it is actually not. Such questionnaires typically ask people if they evacuated and then branch to two different groups of followup items—one group of items for those who did evacuate lists reasons why they did evacuate and different group of items for those who did not evacuate lists reasons why they did not evacuate. As an example, suppose that one reason for not evacuating is "I was concerned about

leaving my pets". The problem with providing this item only for those who did not evacuate is that there are probably people who did evacuate that were also concerned about their pets. Indeed, it is possible that people who did evacuate were just as concerned about their pets as those who did not. Unfortunately, the structure of the questionnaire makes it impossible for the researcher to find that out. A better way to address the issue is to have one item that asks "When you were deciding whether or not to evacuate, to what degree were you concerned about the safety of your pets?" and another item that asks "Did you evacuate?" Calculating the correlation between the responses to these two questions makes it possible to assess the degree to which concern about pets distinguishes between evacuees and non-evacuees rather than assuming that concern for pets is only relevant to non-evacuees.

732 As noted in my comments on line 577, the Baker (1991) narrative review and the Huang et al. (2016) SMA summarize the literature more effectively than any list of individual studies. Additional individual studies are appropriate to include only if they were not included in the Baker (1991) or Huang et al. (2016) reviews. This would be the case for hurricane studies conducted since 2014, or for any studies of inland floods or tsunamis—neither of which were addressed in those reviews.

765 The claim that "most studies focus on past experiences" seems to conflict with the Kellens et al. (2013) statement that only a small amount of research on flood risk perception and communication has studied households' immediate behavioral response to imminent flooding. The apparent discrepancy should be explained.

References Kellens, W., Terpstra, T., & De Maeyer, P. (2013). Perception and communication of flood risks: A systematic review of empirical research. Risk Analysis, 33(1), 24-49. Lindell, M.K. (2018). Communicating imminent risk. In H. Rodríguez, J. Trainor, and W. Donner (eds.) Handbook of Disaster Research, 2nd ed. (pp. 449-477). New York: Springer. Lindell, M.K., Murray-Tuite, P., Wolshon, B. & Baker, E.J. (2019). Large-Scale Evacuation: The Analysis, Modeling, and Management of Emergency Relocation from Hazardous Areas. New York:

Routledge. Lindell, M.K. & Perry, R.W. (2012). The Protective Action Decision Model: Theoretical modifications and additional evidence. Risk Analysis, 32, 616-632. Lindell, M.K., Prater, C.S., Gregg, C.E., Apatu, E., Huang, S-K. & Wu, H-C. (2015). Households' immediate responses to the 2009 Samoa earthquake and tsunami. International Journal of Disaster Risk Reduction, 12, 328-340. Lindell, M.K., Prater, C.S. & Perry, R.W. (2006). Fundamentals of Emergency Management. Emmitsburg MD: Federal Emergency Management Agency Emergency Management Institute. Available at www.training.fema.gov/hiedu/aemrc/booksdownload/fem/ or hrrc.arch.tamu.edu/publications/books%20and%20monographs/ Mileti, D. S., & O'Brien, P. W. (1992). Warnings during disaster: Normalizing communicated risk. Social Problems, 39(1), 40-57. Withey, S.B. (1962). Reaction to uncertain threat. In G.W. Baker and D.W. Chapman (eds.). Man and Society in Disaster (pp. 93-123). New York: Basic Books.

---

## Referee Comment (RC2) · Anonymous Referee #2 · 4 Jul 2020

My concern about this article is the positionning in terms of the potential aid for decision maker, at least the Paris City council services. About this I can see at least 4 weaknesses in the proposed research:

1. The paper is about evacuation capacities of high rise buildings in Paris, in a context of slow flooding process. The river Seine has indeed a slow flooding process and the evidence is that plenty of time is available to evacuate! Why a scientific method is needed for this? Does it really help anyone?

[Figure]

2. Another weakness in the approach choosed is the lack of integration of other services that need to act to protect themselves from floodings. I guess RATP, EDH and others have to act to reduce the vulnerability of their systems (see Serre and Toubin research for example) during this lap of time. These past research works showed that many services, as well as road availability is subject to uncertainties... Why this point is not included here despite its criticity?

3. You tried to define vulnerability profile of inhabitants of such high buildings. Several researches in this area are available and recognised worlwide. Unfortunately these approaches are not even prestented in the paper. For example, how did your proposed evaluation is getting deeper than approaches proposed by Cutter for example?

4. for me, the real problem in such flooding context is not the evacuation process: city managers know how to evacuate cities with million of people. The most important question is: how can we organise the come back home when flooding duration may exceed one month?

To conclude, I do think the reserch proposed does not sound at all with the real need of the City of Paris and Prefecture need in terms of contingency and flood risk management and rescue strategies.

For all these major reasons, I recommend to reject this article proposition.

---

## Author Comment (AC1) · 8 Nov 2020

**Research article: Household resilience to major slow kinetics floods: a prospective survey of the capacity to evacuate in high rise buildings in Paris**

New title: **Household resilience to major slow kinetic floods: a prospective survey of the evacuation capacity in high rise buildings in Paris**

**Nathalie Rabemalanto, Nathalie Pottier, Abla Mimi Edjossan-Sossou, Marc Vuillet**

*Correspondence to : Nathalie Pottier (nathalie.pottier@uvsq.fr), and marc.vuillet@eivp-paris.fr*

| Comments of anonymous Referee # 1 | Response to anonymous Referee # 1 |
|---|---|
| Summary: This manuscript addresses an important and unfortunately neglected issue: flood evacuation from high-rise buildings. The literature review is good but, as ˘ noted below, can be improved. The sample size is excellent, although the response rate is low and so the sample's representativeness is uncertain. However, as the authors note, the response rate is comparable to other mail questionnaires so it is not a major flaw in the study. The cluster analysis of the respondent profiles is a noteworthy innovation. The paper's conclusions … (*no further, cut text …*) | Thank you very much for your detailed and thorough analysis of our research paper. We have improved the bibliography as requested to add references or remove those that are not relevant. We respond below point by point to the comments: |
| • Line 115: Other factors likely to affect flood evacuation decisions are enviC1 NHESSD Interactive comment Printer-friendly version Discussion paper environmental cues; social cues; warning sources, channels, and messages; protective action perceptions, stakeholder perceptions, and personal characteristics (e.g., sensory and physical mobility limitations, hazard experience) see Lindell (2018). Some ˘ of these are mentioned later in the manuscript experience on line 135, warnings on ˘ line 138, social cues on | You are right. We group all the factors likely to affect flood evacuation decisions at the beginning of the text to give the reader a clear overview at the beginning of the article. |

*Nat. Hazards Earth Syst. Sci. Discuss., https://doi.org/10.5194/nhess-2020-150-RC1, 2020*

| | |
|---|---|
| line 147, and environmental cues on line 148. These other factors should be summarized here. | |
| • 128: It is indeed possible that the effects of demographic variables on evacuation is complex. Alternatively, the variation across studies in the significance of demographic variables in predicting evacuation can be explained as little more than random fluctuations that Baker (1991) characterized as small and inconsistent among studies and Huang et al. (2016) Figure I showed have consistently small effect sizes when aggregated in a statistical meta-analysis (SMA). At the very least, the authors should acknowledge that the effect of demographic variables is controversial. | We could write "identifying households likely to evacuate can prove complex and different studies show that the effects of demographic variables is controversial (Baker, 1991; Huang et al., 2016). |
| • 134: I re-read Dash and Gladwin (2007) and I can't find any statements that support the proposition, implicit in this statement, that experience causes risk perception which, in turn, causes evacuation. In fact, Dash and Gladwin repeatedly propose experience and risk perception as competing predictors of evacuation. Also, I can't find the term "awareness" anywhere in the Whitehead et al. (2000) article. | Line 134, we propose to remove the reference to Withehead and the word "awareness", and to reformulate the sentence as follows: "But the decision to leave or to stay for a household does not necessarily depend on its previous experience of disasters. According to Dash and Gladwin (2017), experience and risk perception are not correlated explanatory factors but rather considered to be competitive in terms of evacuation decision". |
| • 232: Huang et al. (2016) cited 11 studies of evacuation expectations (what the present authors call prospective surveys). | I think there is a translation problem in our paper and that we have not translated well the meaning of our thinking through the word "prospective". What we call "prospective" studies in French seems to be named rather "expectations" studies in the United States. Nevertheless, the sense we wanted to give is the same as Huang et al. described in his paper: studies involving expected responses to hypothetical flood scenarios in our case (and in an area where there has never been an evacuation related to a slow large-scale flood).

We affirm that "there are few papers on prospective surveys…" because, on the one hand, our bibliographic research led us to find many more studies involving actual responses to household evacuation than studies involving expected responses to hypothetical evacuation scenarios (whatever the type of natural or technological risk at the origin of the study of households' evacuation); and on the other hand, this is also the observation that reveals the paper of Huang et al. (2016): only 11 studies founded and examined about behavioral expectations studies against 38 identified about post-disaster evacuation |

*Nat. Hazards Earth Syst. Sci. Discuss., https://doi.org/10.5194/nhess-2020-150-RC1, 2020*

| | |
|---|---|
| | studies. It seems to us relatively "few" compared to the abundance of post-disasters studies. |
| • 243 Lindell and Perry (2012) is not listed in the Reference section. | Sorry, we add it in the bibliography. |
| • 286 Social cues are observations of other people's behavior that influence them to evacuate. The most common social cues are observations of businesses closing and other people evacuating. Social assistance is most commonly seen in people staying with peers (friends and relatives) rather than in commercial facilities (hotels or motels), government shelters, or other locations (e.g., second homes). Lindell et al. (2019) Section 6.2 summarized the US research as indicating that an average of 62% of evacuees C2 NHESSD Interactive comment Printer-friendly version Discussion paper stay with peers, 27% stay in commercial facilities, and 3% stay in government shelters (auditoriums or gymnasiums of schools or churches) | Thank you for your detailed precisions. We reformulate the sentence line 286-288: instead of "this is what some authors call the effect of social cues, …", we write: "The question concerning the destination of the "drop-off point" in the event of their building being evacuated, aims to collect information on what some authors call the effects of social cues (observations of other people's behavior that influence them to evacuate). In our case, the responses help to reveal if Parisians expect to receive help from others to be relocated and from whom exactly (from peers, from government help, …). The US research on that point is summarized in Lindell et al. (2019). |
| • 298 The acknowledgement in this statement that some studies especially Baker's (1991) review and the Huang et al. (2016) meta analysis, concluded that demographic variables do not seem to be significant predictors of evacuation needs to be reconciled with the discussion of demographic variables at line 128. | After "varies according to the context", we can add in parentheses: "as mentioned in the second paragraph of part 2). |
| • 337 Huang et al. (2012) also cited a response rate of 24.6% from the Texas coastal evacuation expectations survey by Lindell et al. (2001). In addition, Huang et al. (2012) p. 294 point out that the concern about low response rates is that some groups are under-represented. **However, any bias in demographic characteristics is only relevant to the degree that these characteristics are correlated with evacuation expectation and the variables that are highly correlated with it.** In fact, as noted earlier, the correlations of demographic characteristics with psychological variables and evacuation expectation are generally very small even when they are statistically significant. Consequently, demographic representativeness might not be as big a problem as many authors seem to think. | Thank you for your rigorous analysis, very useful for us. We propose to modify the last paragraph: after the sentence that ends with "that has not been experienced", we add this: So this return rate of 23% for what we call in France "prospective" survey, is relatively similar to previous expectation US surveys (Lindell et al., 2011). Moreover, Huang et al. (2012) point out that the concern of law response rate is not as important as many authors seem to think. It is more linked with the fact that some groups are under-represented than with the fact that demographic characteristics are correlated with psychological variables. Indeed, the correlations of demographic characteristics with psychological variables and evacuation expectation are generally very small even when they are statistically significant (Huang. et al., 2012). |
| • 349 Chang (2009) is mis-cited as Chang (2019) in the Reference section. | Sorry, we will add it in the Reference section. |

*Nat. Hazards Earth Syst. Sci. Discuss., https://doi.org/10.5194/nhess-2020-150-RC1, 2020*

| | |
|---|---|
| • 428 I don't understand what is meant by the occupational category "inactive". The authors should explain this term. | "Inactive" is a person out of work, without employment contract. We add in the text: "inactive (out of work)" |
| • 438 Pets in the home are indeed a likely evacuation impediment, but this obstacle can be avoided if people know that there are places where they can evacuate with their pets. | You are right. We will precise in the text: "This type of person might not be willing to evacuate unless the authorities tell them that there are places where they can evacuate with their pets". |
| • 504 It seems odd that the authors' conclusion about the importance of authorities did not cite the most rigorous and comprehensive review, Huang et al. (2016), because their review shows that this is the most important predictor of evacuation. | You are right. This reference has already been cited several times in our paper but it deserves, here too, to be cited as a priority among the reference studies. We add it and modify the text as follows: "The first reason reflects the same findings as those obtained by Baker (1991), Dash & Gladwin (2007), Kreibich et al. (2017), and the most rigorous and comprehensive review conducted by Huang et al. (2016) on the subject: official warnings are the most important predictor of evacuation decisions." |
| • 544 The discussion in subsequent lines suggests that "responsive" (which implies a passive lack of cooperation) would be a better word choice than "reactive" (which implies active opposition). | Yes, it is a bad translation. We replace the word "reactive" with "responsive". |
| • 546 Although the relationship between how people respond to a behavioral expectations questionnaire and how they actually respond in a disaster is not perfect, it is statistically significant and positive (Kang et al., 2007). Moreover, the variables that predict behavioral expectations also predict actual response in a disaster (Huang et al., 2016). | After "… that one day they could actually be asked to evacuate.", we add your remark in the form of this sentence that supports our argument: "Previous US studies have shown that the variables that predict behavioral expectations also predict actual response in a disaster (Huang et al., 2016). |
| • 577 Baker (1991) reported a narrative review of hurricane evacuation studies conducted prior to his article and Huang et al. (2016) reported a more powerful SMA of hurricane evacuation studies conducted between Baker's review and 2014. The results of the Whitehead et al. (2000) and Dow and Cutter (1998) studies were included in the Huang et al. (2016) SMA, so it is a logical error to cite those two studies as if they provided independent evidence. A rough analogy at the level of a single study would be to find a correlation of r = 0 between gender and evacuation but argue that there is some evidence that women tend to evacuate because a subset of the women did evacuate. Given the similar findings between findings from Baker (1991) and Huang et al. (2016), any studies reporting contrary findings are most likely to be due to random sampling fluctuations. | So we change the sentences. We replace the sentence:

"It was not mentioned earlier in this study because even though some authors, such as Whitehead et al. (2000), found that women were more likely to evacuate, our hypothesis is that gender has no effect on evacuation decisions and capacities, echoing the results of Baker (1991), Dow & Cutter (1998) and Huang et al. (2016) ».

By the sentence:

"Our hypothesis is that gender has no clear effect on evacuation decisions and capacities. International bibliographic analysis, in line with the Baker's results |

*Nat. Hazards Earth Syst. Sci. Discuss., https://doi.org/10.5194/nhess-2020-150-RC1, 2020*

| | |
|---|---|
| | (1991) to the SMA study conducted by Huang et al. (2016), indicates that overall hypothetical evacuation studies, female gender had a moderately consistent percentage nonsignificant." |
| • 625 The conclusion about the relationship between floor level and evacuation expectation would be stronger if supported by a ï¿Ac2 test. | I want to change the paragraph here:

Instead of:

Last, the level of the floor is quite random for most types except, in two cases. In T1, 46% live above the 24th floor, which means that the most autonomous people tend to choose to live on the upper floors. On the contrary, 16 of the 19 people in T4 (not willing to evacuate) live below the 17th floor.

I write:

Last, the relationship between floor level and evacuation decision is quite random for most types except, in two cases. In T1 (people declaring themselves fully autonomous in the event of an evacuation), 46% live above the 24th floor. On the contrary, 16 of the 19 people in T4 (not willing to evacuate) live below the 17th floor. The most autonomous people live on the upper floors while those who are less self-reliant at the time of the survey (even though they moved several years ago) live on the lower floors.
This is rather a positive result for emergency services. But these results reveal, above all, that the people living on the highest floors did not realize the inconvenience associated with the malfunction of the elevators in the choice to leave or stay. We can suppose that it is because they have never experienced this situation. In any case, this data is useful in dispelling misconceptions and improving residents' information about the consequences of power outages (e.g. shutting down elevators) on daily life. It would make it possible to better understand why the authorities recommend high-rise building evacuation before the flood reaches their neighborhood, but from the moment when preventive power cuts are planned. |

*Nat. Hazards Earth Syst. Sci. Discuss., https://doi.org/10.5194/nhess-2020-150-RC1, 2020*

| | |
|---|---|
| • 662 As a minor point, the quote from Dash and Gladwin is true for hurricanes but not necessarily for inland floods—see the Lindell et al. (2019) report on the Uttarakhand ˇ flood. | Yes. We replace the reference Dash and Gladwin (2007) by Lindell et al. (2019) which is more appropriate with regard to the theme of flooding. |
| • 670 The information provided in hazard awareness brochures and hazard awareness meetings is not necessarily limited to those who read the brochures or attend the meetings. Lindell et al. (2015) found that more people knew they should evacuate immediately after a severe earthquake because it could cause a tsunami than had read hazard brochures or attend hazard meetings. The likely reason is that this information was passed through informal social networks either before the earthquake or immediately after it struck. | We can add in line 671: "For post-disasters studies, Lindell et al. (2015) showed that information about passed through informal social networks is more important than information read in brochures or transmitted during the meeting. |
| • 679 The reference to the Colbeau-Justin and de Vanssay (2001) and Becerra et al. (2013) papers raises the question how prevalent were rumors about and denial of the flood risk? Rumors and denial are always present, so the question is whether these were characteristic of 1% of the population or 99% of the population. | Colbeau Justin and de Vanssay explained in their publication that a lack of information for residents affected by the floods of the Somme in 2001 helped spread rumors that the Somme had been flooded to preserve Paris. A detailed information on the flood management of the 2001 Somme event and the kinetic flood process help inhabitants to understand that rumors were wrong. |
| • 681 It is unclear what is meant by "alarm systems". Are these warning systems? | Yes, sorry, this word was poorly translated. We will replace "alarm system" by "warning system". |
| • 694 I think it is fine to credit Thouret and D'Ercole (1996) for presenting the concept of risk personalization but the authors should also mention Mileti and O'Brien (1992), who presented this term earlier based on Withey's (1962, p. 106) theorizing. | Yes, of course, we add this international reference in the text and bibliography. |
| • 702 It is unclear if the authors intend "should not create panic" to mean an injunction ("We don't want this person to create panic because it is possible for that to happen.") or an expectation ("We don't expect that this person will create panic because people won't panic in response to this type of information."). If the authors intended this statement as an injunction, they are mistaken because panic is extremely rare even during life-threatening disasters, see Lindell et al. (2006) Chapter 8. If the authors intended ˇ this statement as an expectation, they should restate it that way. | We propose to precise our results by changing the sentence. Instead of "Anyway, the person who determines and shares such information should not create panic among the population while informing them about flood risk". The new sentence will be: "It is on this point that the information of residents must progress in an educational and non-anxiety-provoking manner. It will increase the proportion of preventive autonomous evacuation of households and will thus facilitate the local management of flooding by the authorities". |

*Nat. Hazards Earth Syst. Sci. Discuss., https://doi.org/10.5194/nhess-2020-150-RC1, 2020*

| | |
|---|---|
| • 716 As noted in my comments on line 546, there is relevant research on the relationship between expected and actual evacuation behavior. | You are right, here I have to remind their results, so I change the sentences.

Instead of:
However, even being well informed does not entirely guarantee that the real action would be the same as the one mentioned in the completed questionnaire. Nevertheless, the descriptive statistics showed some particularly coherent answers, for example for T1 (totally autonomous), T2a (partially dependent regarding the relocation place and/or the means of transport to get there) or T3 (totally dependent).

The new sentence will be: "Relevant research summarized in the paper by Huang and al. (2016), prove a notable relationship between how people respond to a behavioral expectations questionnaire and how they actually respond in a disaster. Therefore these prospective surveys (also called in the USA expectation surveys) deserve to be developed in hazardous areas France. |
| • 726 Asking people to endorse specific reasons why they didn't evacuate seems like a good idea, but it is actually not. Such questionnaires typically ask people if they evacuated and then branch to two different groups of follow up items one group of ˘ items for those who did evacuate lists reasons why they did evacuate and different group of items for those who did not evacuate lists reasons why they did not evacuate. As an example, suppose that one reason for not evacuating is "I was concerned about C5 NHESSD Interactive comment Printer-friendly version Discussion paper leaving my pets". The problem with providing this item only for those who did not evacuate is that there are probably people who did evacuate that were also concerned about their pets. Indeed, it is possible that people who did evacuate were just as concerned about their pets as those who did not. Unfortunately, the structure of the questionnaire makes it impossible for the researcher to find that out. A better way to address the issue is to have one item that asks "When you were deciding whether or not to evacuate, to what degree were you concerned about the safety of your pets?" and another item that asks "Did you evacuate?" Calculating the correlation between the responses to these two questions makes it possible to assess | Thank you for your comment. We will take it into account to improve the questionnaire for further surveys. |

*Nat. Hazards Earth Syst. Sci. Discuss., https://doi.org/10.5194/nhess-2020-150-RC1, 2020*

| | | |
|---|---|---|
| | the degree to which concern about pets distinguishes between evacuees and non-evacuees rather than assuming that concern for pets is only relevant to non-evacuees. | |
| • | 732 As noted in my comments on line 577, the Baker (1991) narrative review and the Huang et al. (2016) SMA summarize the literature more effectively than any list of individual studies. Additional individual studies are appropriate to include only if they were not included in the Baker (1991) or Huang et al. (2016) reviews. This would be the case for hurricane studies conducted since 2014, or for any studies of inland floods or tsunamisâA˘Tneither of which were addressed in those reviews. | We replace: (Baker, 1991; Gladwin et al. 734 2001; Huang et al., 2012; Riad et al., 2006; Lindell et al., 2005; Whitehead et al., 2000)

By :
The Baker narrative review (Baker, 1991) of hurricane evacuation studies and the Huang and al. (2016) statistical meta-analyses summarize the literature on this topic. |
| • | 765 The claim that "most studies focus on past experiences" seems to conflict with the Kellens et al. (2013) statement that only a small amount of research on flood risk perception and communication has studied households' immediate behavioral response to imminent flooding. The apparent discrepancy should be explained. | We wrote: "it is a study dealing with anticipation, while most studies focus on past experiences".

We change in: "it is a study dealing with anticipation of household behavior while most studies focus on actual responses to past flood events, especially in France (what we call what we call REX our RETEX for "experience feedback analysis"). In addition, the issue of evacuating floods from high-rise buildings in metropolitan areas subject to flooding is a question that is little or not addressed (in France). |

*Nat. Hazards Earth Syst. Sci. Discuss., https://doi.org/10.5194/nhess-2020-150-RC1, 2020*

---

## Author Comment (AC2) · 8 Nov 2020

**Research article: Household resilience to major slow kinetics floods: a prospective survey of the capacity to evacuate in high rise buildings in Paris**

New title: **Household resilience to major slow kinetic floods: a prospective survey of the evacuation capacity in high rise buildings in Paris**

**Nathalie Rabemalanto, Nathalie Pottier, Abla Mimi Edjossan-Sossou, Marc Vuillet**

*Correspondence to : Nathalie Pottier ([nathalie.pottier@uvsq.fr](mailto:nathalie.pottier@uvsq.fr)), and [marc.vuillet@eivp-paris.fr](mailto:marc.vuillet@eivp-paris.fr)*

| Comments of anonymous Referee # 2 | Response to anonymous Referee # 2 |
|---|---|
| My concern about this article is the positionning in terms of the potential aid for decision maker, at least the Paris City council services. About this I can see at least 4 weaknesses in the proposed research: | Thank you for taking the time to review our article. I have no doubt that we will be able to respond to the comments made. |
| • 1. The paper is about evacuation capacities of high rise buildings in Paris, in a context of slow flooding process. The river Seine has indeed a slow flooding process and the evidence is that plenty of time is available to evacuate! Why a scientific method is needed for this? Does it really help anayone? | You are right, this is a slow-kinetic flood. However, we wonder about the risk that unexpected water inflows and / or shutdowns (as was the case in the RERC in 2016) triggering sudden self-evacuation phenomena, difficult to predict, complicating management. crisis. Also, whatever the time allotted, the crisis management services of Paris Police Prefecture (Paris Defense and Security Zone) expressed at the end of crisis simulation exercises we attended, their need to know about the capacity of people to self-evacuate, and, above all, to self-evacuate. relocate, where to relocate, and if it is then possible for them to continue working and / or telework. We probably haven't made that clear. Also we added a sentence in this direction line 54

We added line 54 to 59 : Above all, there remains a lot of uncertainty regarding the potential zoning of floods and the effects on the functioning of networks (Gache, 2014). The crisis management services fear that very few inhabitants will agree to self-evacuate as long as the effects of the flood remain barely noticeable, and that, suddenly, following the shutdown of certain services and / or unexpected water inflows (as was the case with the RERC or in the 16th |

*Nat. Hazards Earth Syst. Sci. Discuss., https://doi.org/10.5194/nhess-2020-150-RC2*

| | |
|---|---|
| | district of Paris during the 2016 flood), a large number of people suddenly need to be taken care of (Edjossan-Sossou et al., 2020). |
| • 2. Another weakness in the approach choosed is the lack of integration of other services that need to act to protect themselves from floodings. I guess RATP, EDH and others have to act to reduce the vulnerability of their systems (see Serre and Toubin research for example) during this lap of time. These past research works showed that many services, as well as road availability is subject to uncertainties... Why this point is not included here despite its criticity? | Absolutely, the issue of networks is central and this is why we cite the work of Toubin et al., 2015 and Bocquentin et al 2020 in our introduction. Also, during our discussions, as part of a flood simulation exercise, the authorities of the Ile-de-France region (prefecture and city) very clearly communicated their questions to us concerning the number of people that should be taken care of. , especially at what times, according to the reactions to the instructions, the influence of the flooded surfaces and the shutdown of the networks. This last point being very uncertain, for example the REC was cut in 2016 and 2018 well before reaching the water level foreseen in the crisis management plan. Above all, the business continuity plans of network managers are constantly evolving, depending on the actions taken to reduce vulnerability, or conversely the discovery of new vulnerabilities (as was the case for the RERC in 2016). The objective is to produce information concerning the populations exposed to the risk and their behavior, regardless of the flood scenario and network interruption. We aded line 83 to 88 :Especially the knowledge on the zoning of floods and the business continuity plans of the network managers is constantly evolving, in particular according to the actions taken to reduce the vulnerability of the infrastructure and / or the discovery of new vulnerabilities (as was the case for the RERC in 2016). Thus, the objective is to produce information on the populations exposed to the risk and their behaviors that can be exploited regardless of the flood scenario and network interruption. |
| • 3. You tried to define vulnerability profile of inhabitants of such high buildings. Several researches in this area are available and recognised worlwide. Unfortunately these approaches are not even prestented in the paper. For example, how did your proposed evaluation is getting deeper than approaches proposed by Cutter for example? | Indeed, many authors have produced very interesting research concerning the vulnerability profiles of populations exposed to risks. Thus Cutter et al, 2003, who proposes a global social vulnerability index, often cited in references in our article. However, we do not seek to measure vulnerability profiles strictly speaking, but, more precisely, to quantify the ability to self-evacuate and to self-relocate. Thanks for your comment, we may have been imprecise, so we added lines 122 to 133: Several authors, such as Cutter et al. (2003) propose global social resilience indicators, at the level of regions or metropolitan areas (Cutter et al., 2014). This type of approach can provide very interesting information to give orders of magnitude aiding in crisis management planning. It enables the study of hazard zoning, exposure and assessment of the overall social vulnerability of |

*Nat. Hazards Earth Syst. Sci. Discuss., https://doi.org/10.5194/nhess-2020-150-RC2*

| | populations. We mentioned in the introduction an approach of this type implemented in the Paris metropolitan area (Fujiki & Renard, 2018). Using statistics, particularly social statistics, for the territory, it provides an overview of the evacuation problem. In addition to the results obtained by Fujiki & Renard (2018), responding to concerns expressed by crisis managers, the prefecture (civil security) and the city of Paris, we wish here to provide some more detailed answers concerning, in particular, the factors likely to cause people to evacuate, their capacity to self-evacuate, the location of their self-hosting and whether it is possible for them to continue their professional activity.

So we have added the references in our bibliography:
19. Cutter, S. L., B. J. Boruff and L. W Shirley, Social Vulnerability to Environmental Hazards, Social Science Quarterly, 2003, vol. 84, n ° 1, pp. 242-261.
DOI: 10.1111 / 1540-6237.8402002
18.20. Cutter, S.L., R. Schumann, and C.T. Emrich. "Exposure, Social Vulnerability and Recovery Disparities in New Jersey after Hurricane Sandy", Journal of Extreme Events, 2014. 1 (1): 23 p |
|---|---|
| • 4. for me, the real problem in such flooding context is not the evacuation process: city managers know how to evacuate cities with million of people. The most important question is: how can we organise the come back home when flooding duration may exceed one month? To conclude, I do think the reserch proposed does not sound at all with the real need of the City of Paris and Prefecture need in terms of contingency and flood risk management and rescue strategies. For all these major reasons, I recommend to reject this article proposition. | I do not understand this remark, since the motivation for the work carried out comes from a discussion with the crisis management services of the Parisian territory: Paris Police Prefecture (Paris Defense and Security Zone), crisis management services of the city of Paris and the 15th arrondissement.
See section Acknowledgments
As we indicated in response to the previous questions, the authorities (Prefecture, City of Paris and town hall of the 15th district of Paris) told us of their concern to better understand the behavior of households likely to self-evacuate. We have therefore prioritized this point. In perspective, it may indeed be interesting to also study the subject of returning to their home during the recession phase, but this is the subject of another study. Note that we included in our survey elements concerning the capacity of the populations to work from the place where they have relocated, which could influence their time to return to their homes. Also, we added as a perspective the interest of such a study.
We have added line 304-306: In perspective, it would also be interesting to study the phase of return to normalcy, and the factors likely to influence people to make the decision to return home. |

*Nat. Hazards Earth Syst. Sci. Discuss., https://doi.org/10.5194/nhess-2020-150-RC2*

*Nat. Hazards Earth Syst. Sci. Discuss., https://doi.org/10.5194/nhess-2020-150-RC2*